# A signal motif retains Arabidopsis ER-α-mannosidase I in the *cis*-Golgi and prevents enhanced glycoprotein ERAD

Jennifer Schoberer [1], Julia König[1], Christiane Veit[1], Ulrike Vavra[1], Eva Liebminger[1], Stanley W. Botchway [2], Friedrich Altmann [3], Verena Kriechbaumer [4], Chris Hawes [4] & Richard Strasser [1]

The Arabidopsis ER-α-mannosidase I (MNS3) generates an oligomannosidic N-glycan structure that is characteristically found on ER-resident glycoproteins. The enzyme itself has so far not been detected in the ER. Here, we provide evidence that in plants MNS3 exclusively resides in the Golgi apparatus at steady-state. Notably, MNS3 remains on dispersed punctate structures when subjected to different approaches that commonly result in the relocation of Golgi enzymes to the ER. Responsible for this rare behavior is an amino acid signal motif (LPYS) within the cytoplasmic tail of MNS3 that acts as a specific Golgi retention signal. This retention is a means to spatially separate MNS3 from ER-localized mannose trimming steps that generate the glycan signal required for flagging terminally misfolded glycoproteins for ERAD. The physiological importance of the very specific MNS3 localization is demonstrated here by means of a structurally impaired variant of the brassinosteroid receptor BRASSI-NOSTEROID INSENSITIVE 1.

---

[1] Department of Applied Genetics and Cell Biology, University of Natural Resources and Life Sciences, Vienna, Muthgasse 18, 1190 Vienna, Austria. [2] Central Laser Facility, Science and Technology Facilities Council (STFC), Rutherford Appleton Laboratory, Research Complex at Harwell, Didcot OX11 0QX, UK. [3] Department of Chemistry, University of Natural Resources and Life Sciences, Vienna, Muthgasse 18, 1190 Vienna, Austria. [4] Department of Biological and Medical Sciences, Faculty of Health and Life Sciences, Oxford Brookes University, Gipsy Lane, Headington, Oxford OX3 0BP, UK. Correspondence and requests for materials should be addressed to J.S. (email: jennifer.schoberer@boku.ac.at)

Asparagine (N)-linked protein glycosylation is the most prevalent form of glycosylation in all eukaryotes. The correct and efficient biosynthesis and processing of N-linked glycans is crucial for all organisms, as N-glycans play important roles in many biological processes, including protein folding, glycan-dependent quality control, protein stability, and protein–protein interactions[1–3]. The process of N-glycosylation is initiated in the ER when the oligosaccharyltransferase complex catalyzes the en bloc transfer of the preassembled oligosaccharide precursor $Glc_3Man_9GlcNAc_2$ to selected asparagine residues in the sequence Asn-X-Ser/Thr (where X can be any amino acid except Pro) within nascent polypeptide chains[4]. N-glycans are then subjected to processing by a cohort of ER- and Golgi-resident glycosyltransferases and glycosidases that act sequentially in a highly coordinated fashion. In the first step, glucose residues are cleaved off from the N-glycan by α-glucosidase I (GCSI) and α-glucosidase II (GCSII, Fig. 1a), which initiates N-glycan-dependent quality control mechanisms in the ER. The best-characterized processes are the calnexin/calreticulin cycle and the ER-associated degradation (ERAD) pathway, where specific N-glycan moieties and ER lectins are involved in facilitating proper folding of newly formed glycoproteins[5]. The resulting $Man_9GlcNAc_2$ oligosaccharide is further processed by the class I α-mannosidases ER-α-mannosidase I (MNS3) and Golgi-α-mannosidase I (MNS1/2) that collectively remove four α1,2-linked mannose residues from the N-glycan. The resulting $Man_5GlcNAc_2$ N-glycan is the final product of early N-glycan

processing steps and forms the substrate for the formation of hybrid and complex N-glycans in the Golgi apparatus[6,7].

The importance of class I α-mannosidases in N-glycan processing of secretory glycoproteins has been recognized recently in plants due to the identification of α-mannosidase mutants (*mns*) impaired in demannosylation of N-glycans. In *Arabidopsis thaliana*, mutants deficient in MNS1–MNS3 (*mns1 mns2 mns3*) accumulate high amounts of oligomannosidic N-glycans without any further processing to complex N-glycans and display defects in root development and cell wall biosynthesis[7]. MNS3 fulfils the biosynthetic role of an ER-α-mannosidase by removing a single terminal mannose residue from the middle branch (B-branch) of the $Man_9GlcNAc_2$ oligosaccharide forming the $Man_8GlcNAc_2$ isomer B (Fig. 1a). MNS1 and MNS2 are two functionally redundant Golgi-α-mannosidases that act downstream of MNS3 in the N-glycan processing pathway, and utilize the $Man_8GlcNAc_2$ substrate to generate $Man_5GlcNAc_2$. The other two Arabidopsis class I α-mannosidases MNS4 and MNS5 are not involved in regular N-glycan processing of properly folded glycoproteins and instead generate a unique N-glycan structure in the ER that leads to the ERAD of misfolded variants of the heavily glycosylated brassinosteroid receptor BRASSINOSTEROID INSENSITIVE 1 (BRI1)[8].

The MNS3-catalyzed formation of $Man_8GlcNAc_2$ is the last N-glycan processing step that is conserved in yeast, mammals, and plants. In plants, ER-resident glycoproteins contain large amounts of $Man_8GlcNAc_2$ oligosaccharides[9–11], indicating that

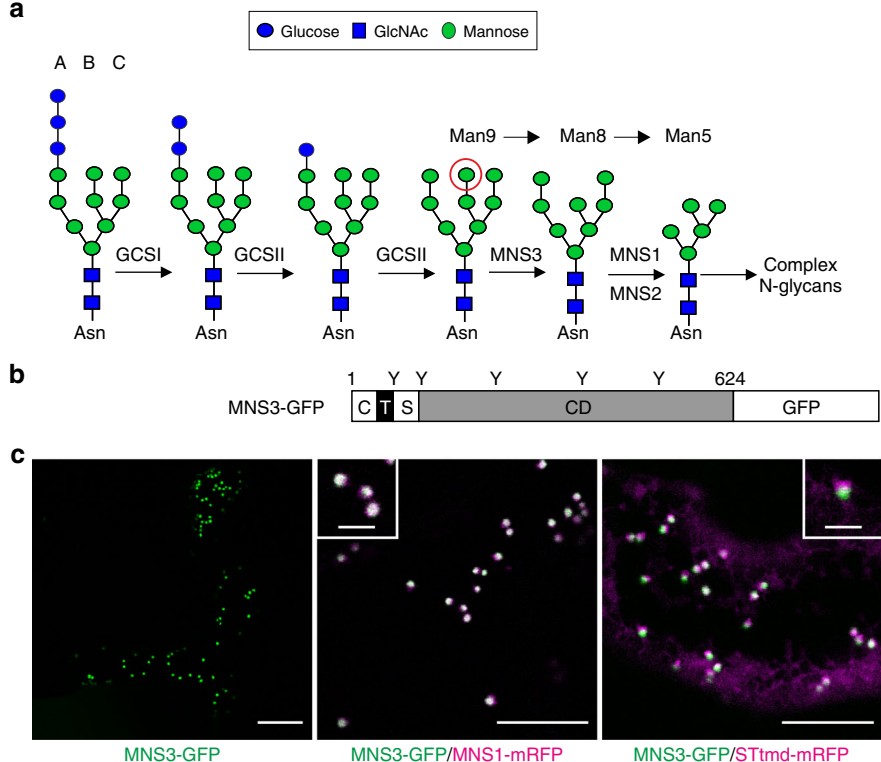

**Fig. 1** Arabidopsis MNS3 is a Golgi-resident protein. **a** The early steps of N-glycan processing in *A. thaliana* are shown. The three branches of the N-linked $Glc_3Man_9GlcNAc_2$ oligosaccharide precursor are marked A, B, and C. The red circle indicates the terminal α1,2-linked mannose residue that is cleaved off by MNS3 from the B-branch. GCSI: α-glucosidase I; GCSII: α-glucosidase II; MNS3: ER-α-mannosidase I; MNS1/MNS2: Golgi-α-mannosidase I. **b** Illustration of the domain organization of MNS3. C, cytoplasmic tail (41 aa); T, predicted transmembrane domain (26 aa); S, stem region (42 aa); CD, catalytic domain (515 aa). Positions of putative N-glycosylation sites are indicated (Y). The depicted numbers represent the length of the protein (624 amino acids). **c** Subcellular localization of MNS3. Confocal images show *N. benthamiana* leaf epidermal cells transiently expressing MNS3-GFP (green) alone (scale bar = 20 μm) and in combination with the *cis*/medial-Golgi protein MNS1-mRFP (magenta, scale bar = 5 μm) or the medial/*trans*-Golgi marker STtmd-mRFP (magenta, scale bar = 5 μm). The insets show a higher magnification of individual dual-colored Golgi stacks (scale bar = 2 μm). Images were acquired 2 days post infiltration (dpi)

these proteins have been processed by MNS3, the only known plant α-mannosidase that can efficiently hydrolyze the respective mannose residue in plants. Importantly, $Man_8GlcNAc_2$ structures on ER-resident glycoproteins are not further processed into $Man_5GlcNAc_2$, the oligosaccharide structure that is generated by Golgi-resident MNS1/2, which suggests a spatial separation of the biosynthetic activities of MNS3 in the ER and MNS1/2 in the Golgi. MNS3, however, does not display any ER location. During fluorescence imaging of *Nicotiana benthamiana* or *A. thaliana* leaf epidermal cells, a fusion of MNS3 to GFP (MNS3–GFP[7]) was sighted in disperse and motile punctate structures strongly reminiscent of Golgi stacks that were also labeled with β1,2-*N*-acetylglucosaminyltransferase I (GnTI), a well-characterized N-glycan processing enzyme residing in *cis*/medial-Golgi membranes[7,11]. Due to the compact organization of the ER-Golgi interface in higher plants, such a punctate fluorescence pattern could equally point at a concentration at ER export sites (ERES), which are enriched in coat protein complex II[12,13].

A putative ER localization of MNS3 has been initially inferred from the yeast ortholog Mns1p that was found to reside in the ER[14,15]. In contrast, reports on the subcellular whereabouts of the mammalian ortholog have been inconsistent. Human ERManI was localized to various sites such as the ER[16,17], the pericentriolar ER-derived quality control compartment[18], and QC vesicles[19] or the Golgi apparatus[20,21]. In addition to their N-glycan processing function, the mammalian and yeast MNS3 orthologs also play a central role in ER-mediated protein QC and glycoprotein ERAD. Plants have a similar ERAD pathway[22], but no particular role in the ERAD of plant glycoproteins has so far been ascribed to MNS3[8], and neither QC vesicles nor distinct QC compartments have been identified in plants.

Altogether, this raises the question of where Arabidopsis MNS3 is located at steady state, and which underlying mechanisms determine this peculiar subcellular localization and its biological significance. Herein, we provide evidence that (1) MNS3 concentrates in *cis*-Golgi membranes at steady state, (2) a tetrapeptide signal within the cytoplasmic tail of MNS3 dictates its retention at the Golgi, and (3) its Golgi retention is a means to spatially separate MNS3-mediated N-glycan processing steps from those that initiate the formation of the ERAD signals on misfolded glycoproteins in the ER.

## Results

**MNS3 concentrates in early-Golgi cisternae.** MNS3 is a membrane-bound protein with an N-terminal cytoplasmic tail (C), a single transmembrane domain (T), and a luminal stem (S) that is linked to the large catalytic domain (CD), which is typical for ER- and Golgi-resident glycosyltransferases and glycosidases (Fig. 1b). It was shown for MNS3 that its N-terminal CTS region confers proper subcellular targeting[7]. To reexamine the subcellular localization of full-length MNS3, MNS3-GFP[7] was transiently expressed in *N. benthamiana* leaf epidermal cells using Agrobacterium-mediated transformation[23] and analyzed using confocal laser scanning microscopy. In agreement with previous live-cell imaging experiments, MNS3-GFP concentrated on disperse and motile punctate structures reminiscent of Golgi stacks (Fig. 1c). Co-expression of MNS3-GFP with MNS1 fused to mRFP (MNS1–mRFP[24]), which acts downstream of MNS3 in the N-glycan processing pathway and resides in *cis*/medial-Golgi cisternae[25,26], showed the co-localization of both proteins, whereas only a fractional overlap with the signal of the non-plant medial/*trans*-Golgi membrane marker STtmd-mRFP[27,28] (52 N-terminal amino acids of the CTS region of rat α2,6-sialyltransferase fused to mRFP) was observed (Fig. 1c). The degree of co-localization in Golgi stacks was assessed by calculating the

Pearson's correlation coefficient ($r$). Our co-localization analyses revealed a positive correlation between MNS3-GFP and MNS1-mRFP ($r = 0.82 \pm 0.07$, mean ± s.d., *T*-test statistic, $n = 12$ images from two biological replicates), which was significantly higher than that for MNS3-GFP co-expressed with STtmd-mRFP ($r = 0.63 \pm 0.09$, $n = 9$ images from two biological replicates, $p < 0.0001$). There was no evidence for the presence of MNS3 in the ER. This result strongly indicates that MNS3 concentrates predominantly in *cis*/medial-Golgi cisternae.

To investigate the localization of MNS3 under native conditions, we expressed MNS3-GFP under the control of its own promoter in *A. thaliana mns3* knockout plants[7]. The MNS3::MNS3-GFP construct fully complemented the N-glycan processing defect of the MNS3-deficient plants (Fig. 2a) and labeled Golgi stacks, which was confirmed by co-localization with the *bona fide* Golgi marker AtGnTI-mRFP[29,30] (Fig. 2b).

The MNS3 amino acid sequence harbors five potential N-glycosylation sites (Asn-69, Asn-114, Asn-236, Asn-377, and Asn-503, Fig. 1b). Since N-glycan structures can serve as markers for the intracellular transport or localization of proteins, we hence determined the glycosylation state of MNS3. ER-localized proteins typically carry incompletely processed oligomannosidic N-glycan structures (like $Man_8GlcNAc_2$), while forward movement to the Golgi will lead to processing of N-glycans and the formation of complex N-glycans carrying β1,2-xylose and core α1,3-fucose residues (main glycoform: $GlcNAc_2Man_3XylFucGlcNAc_2$ = GnGnXF). To discriminate between the two states, total protein extracts from *A. thaliana* (ecotype Col-0) wild-type and mutant plants that lack core α1,3-fucosyltransferase activity (*fut11 fut12*[31]) were subjected to endoglycosidase H (Endo H) and peptide-N-glycosidase F (PNGase F) digestions. Endo H removes oligomannosidic N-glycans, but not Golgi-processed ones, whereas PNGase F cleaves oligomannosidic and complex N-glycans, except those carrying core α1,3-fucose. SDS-PAGE separation and immunoblotting with antibodies against endogenous MNS3 revealed a clear electrophoretic mobility shift of PNGase F-treated MNS3 in the absence of α1,3-fucose, which shows that MNS3 is glycosylated (Fig. 2c). Notably, the absence of a mobility shift of PNGase F-treated MNS3 isolated from wild-type plants that still produce complex N-glycans with core α1,3-fucose residues provides further evidence that MNS3 is a dynamic protein that is able to reach medial-to-*trans* Golgi cisternae where α1,3-fucosylation typically takes place. For a detailed characterization of the MNS3 N-glycosylation profile, MNS3-GFP was transiently expressed in wild-type *N. benthamiana* leaves and affinity-purified. Purified MNS3-GFP was trypsin-digested and subjected to liquid chromatography–electrospray ionization–mass spectrometry (LC–ESI–MS). The predominant N-glycan was found to be the complex N-glycan structure GnGnXF (Fig. 2d). All in all, the obtained data indicate that MNS3 is predominantly located in early-Golgi membranes at steady-state and carries Golgi-processed complex N-glycans.

**MNS3 remains on punctate structures upon Golgi disassembly.** The fungal metabolite brefeldin A (BFA) is a classic tool to study the dynamics of Golgi membranes and its associated proteins. BFA reversibly blocks secretion in mammals, yeast, and plants and inhibits the assembly of COPI required for retrograde intra-Golgi and Golgi-to-ER transport, which typically leads to Golgi stack disassembly and redistribution of Golgi membrane markers into the ER[32,33]. To examine the response of Golgi-resident MNS3 to BFA, we transiently expressed MNS3-GFP in *N. benthamiana* leaves that were subsequently treated with BFA ($100 \mu g\ ml^{-1}$) for 2 h. Surprisingly, MNS3-GFP mainly labelled well-distributed, mobile punctate structures in BFA-treated cells,

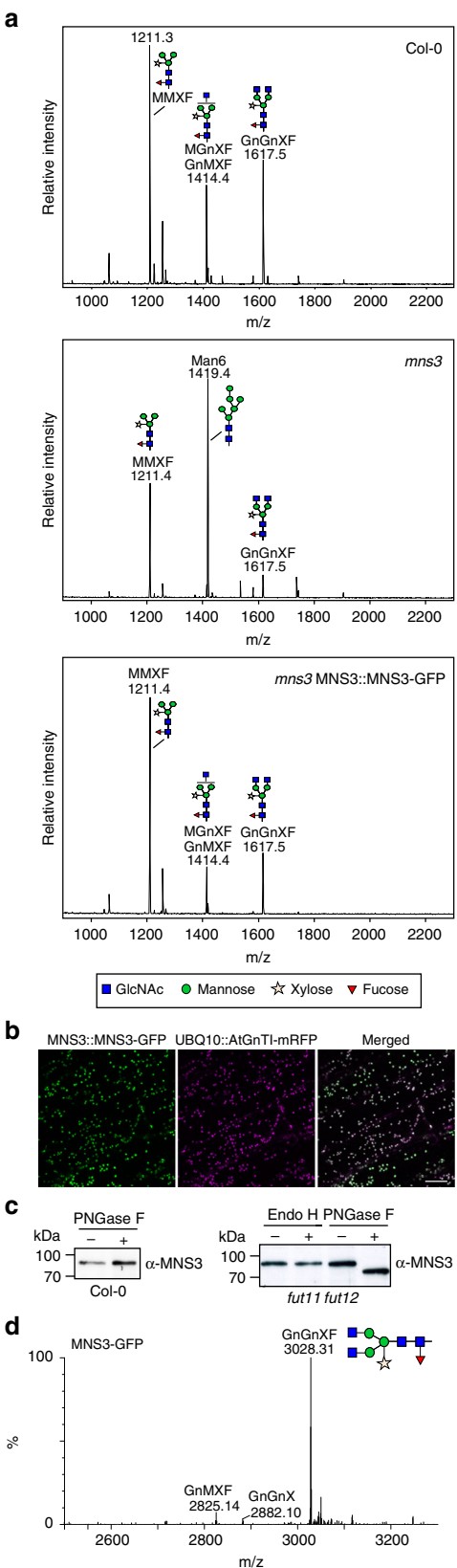

**Fig. 2** MNS3 carries Golgi-processed complex N-glycans. **a** Matrix-assisted laser desorption ionization time-of-flight (MALDI-TOF) mass spectra of total N-glycans extracted from leaves of Arabidopsis wild-type (Col-0) plants, *mns3*, and *mns3* complemented with MNS3::MNS3-GFP. **b** Confocal images of transgenic Arabidopsis *mns3* seedlings expressing MNS3::MNS3-GFP (green) crossed with *gntl* knockout plants expressing UBQ10::AtGnTI-mRFP[29] (magenta). Scale bar = 10 μm. **c** Endo H and PNGase F digestion of crude protein extracts from Arabidopsis wild-type (Col-0) and mutant plants lacking core α1,3-fucosyltransferase activity (*fut11 fut12*). Proteins were separated by SDS-PAGE and blots were probed with an anti-MNS3 antibody. Source data are provided as a Source Data file. **d** Liquid chromatography–electrospray ionization–mass spectrometry (LC-ESI-MS) of MNS3-GFP transiently expressed in *N. benthamiana* leaves. The mass spectrum of glycosylation site 2 (GSST**NGS**TISNSDPK) is shown

complete redistribution into the ER (Fig. 3b). MNS3-labelled puncta appeared closely associated with the ER. The localization pattern of both fusion proteins did not change after longer BFA incubation times or by the application of different BFA concentrations (Supplementary Fig. 1).

The redistribution of Golgi membrane proteins to the ER can also be achieved by the transient expression of a GTP-locked version of the small GTPase SAR1, which inhibits cargo exit from the ER through disruption of the COPII coat complex at ERES[34,35]. To test whether MNS3 also remained on puncta after blocking ER-to-Golgi transport, we transiently co-expressed MNS3-GFP and STtmd-mRFP with SAR1-GTP. Similar to the BFA experiment, MNS3-GFP was mainly sighted in puncta (and very weakly in the ER at high expression levels), whereas STtmd-mRFP was fully redistributed to the ER (Fig. 3b).

The response of MNS3 to BFA treatment and SAR1-GTP expression, respectively, was strikingly similar to that of the golgin AtCASP, a well-studied *cis*-Golgi membrane protein that appears to be involved in an early stage of Golgi stack reformation and possibly at the level of ERES[35,36]. It was previously shown that when secretion was blocked by BFA or SAR1-GTP expression in tobacco leaf epidermal cells, GFP-AtCASP continued to label puncta, which colocated with the ERES marker SAR1–GTP–YFP, whereas co-expressed STtmd-mRFP was fully redistributed to the ER[35]. To examine whether MNS3-labelled punctate structures coincided with AtCASP upon Golgi disassembly, cells co-expressing MNS3-GFP and mRFP-AtCASP were either treated with BFA or SAR1-GTP was transiently co-expressed. In either case, the fusion proteins largely colocated on well-distributed punctate structures. mRFP-AtCASP also clearly labelled the ER, whereas MNS3-GFP highlighted the ER rather weakly (Fig. 3c). This weak ER labelling was also observed after BFA treatment of cells co-expressing MNS3-GFP with the mRFP-tagged ER-resident protein DAD1[37] (Defender against Apoptotic Death-1, Supplementary Fig. 8). We think that upon Golgi membrane disruption both MNS3 and AtCASP accumulate at puncta that may represent membrane structures arising from an incomplete disassembly of Golgi stacks[38].

**Distinct *cis*-Golgi proteins respond differently to BFA**. Besides AtCASP, it was recently shown that the *cis*-Golgi membrane protein AtSYP31[39], a Qa-SNARE required for anterograde ER-to-Golgi traffic, displays a strikingly similar behavior as MNS3 in the presence of BFA[40]. GFP-tagged AtSYP31 (GFP-AtSYP31) was described to accumulate on small punctate structures termed "Golgi entry core compartment (GECCO)" that was suggested to contain a particular subset of *cis*-Golgi proteins and form in the vicinity of ERES upon BFA treatment of tobacco BY-2 cells. GECCO was proposed to act as a scaffold for Golgi reformation

with ER being labelled only very weakly (Fig. 3a). The observed puncta varied in size; the larger ones, however, were strongly reminiscent of Golgi stacks. In BFA-treated cells co-expressing MNS3-GFP and STtmd-mRFP, a large fraction of MNS3-GFP continued to concentrate on punctate structures after 2 h, whereas STtmd-mRFP clearly labelled the ER network, indicating

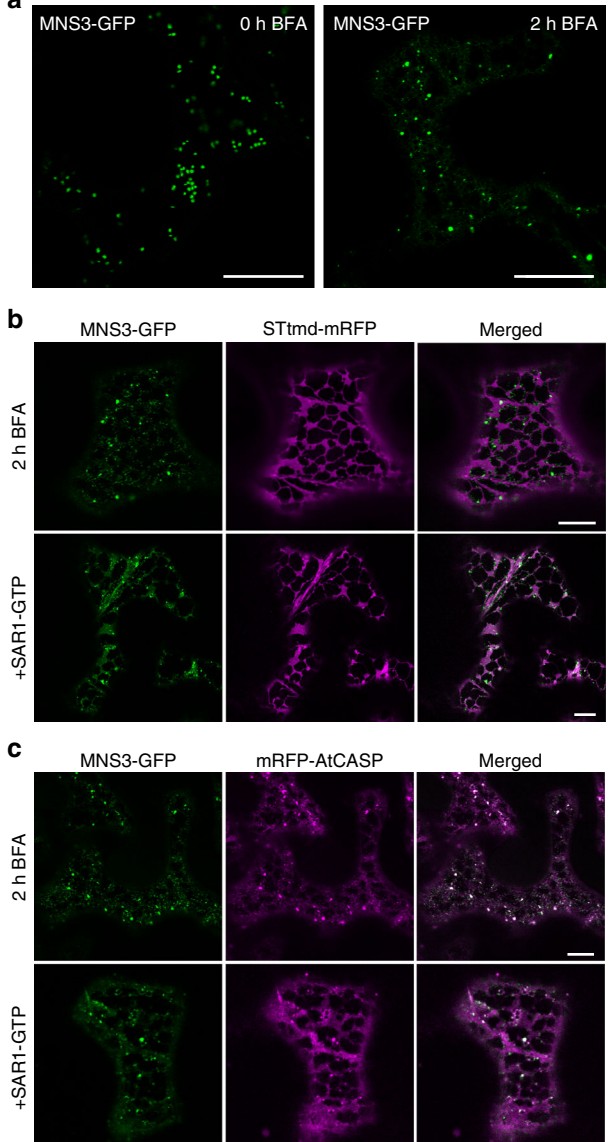

**Fig. 3** MNS3 largely remains on punctate structures during Golgi disassembly. Fusion proteins were transiently expressed in *N. benthamiana* leaf epidermal cells and observed 2–3 dpi using confocal microscopy. **a** Confocal images showing the subcellular localization of MNS3-GFP before (0 h BFA) and after BFA addition (2 h BFA). Scale bars = 20 μm. **b** A representative cell that co-expresses MNS3-GFP (green) with STtmd-mRFP (magenta) and was treated with BFA for 2 h (upper panel) or additionally expresses GTP-locked SAR1 (SAR1-GTP, lower panel). Scale bars = 10 μm. **c** A representative cell that co-expresses MNS3-GFP (green) with the *cis*-Golgi golgin mRFP-AtCASP (magenta) and was treated with BFA for 2 h (upper panel) or additionally expresses SAR1-GTP (lower panel). Scale bars = 10 μm

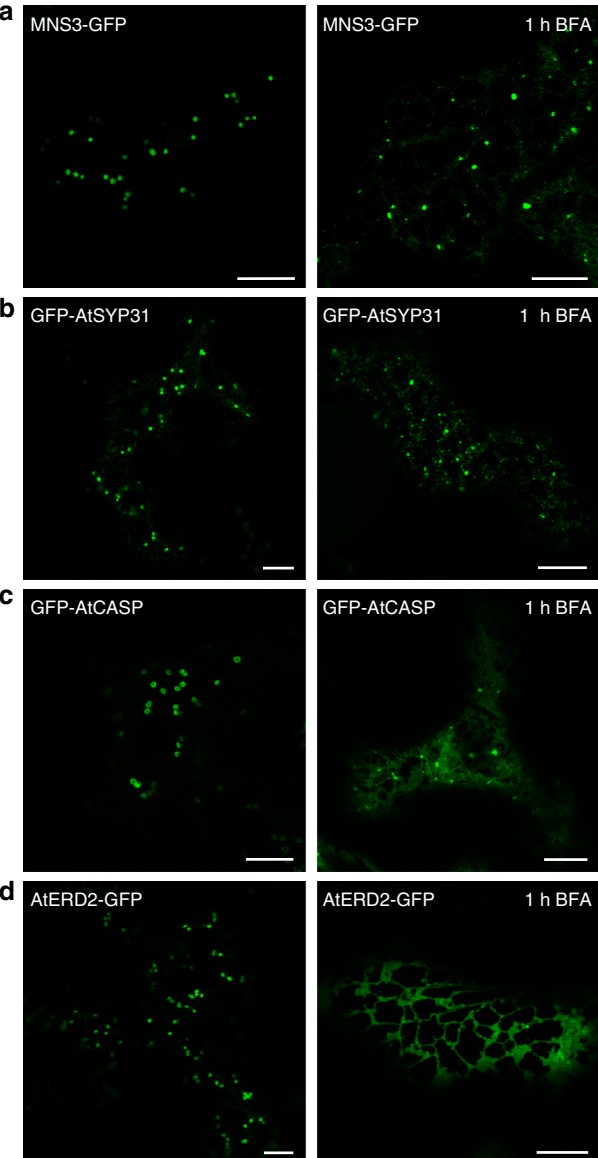

**Fig. 4** Distinct *cis*-Golgi membrane proteins respond differently to BFA. Fluorescent protein fusions were transiently expressed in *N. benthamiana* leaf epidermal cells and observed 2 dpi on a confocal microscope. Images show the subcellular localization of (**a**) MNS3-GFP, (**b**) the Qa-SNARE GFP-AtSYP31, (**c**) the golgin GFP-AtCASP, and (**d**) the K/HDEL recycling receptor AtERD2-GFP before and after 1 h BFA treatment. Scale bars = 10 μm

after BFA removal by receiving Golgi material from the ER[40,41]. To compare the BFA-induced localization pattern of MNS3 with that of AtSYP31, we expressed MNS3-GFP and GFP-AtSYP31, respectively, in *N. benthamiana* leaves and observed their response to BFA via confocal microscopy. Before BFA addition, both fusion proteins exclusively concentrated on Golgi stacks (Fig. 4a, b). One hour after BFA addition, MNS3-GFP and GFP-AtSYP31 highlighted a dispersed pattern of puncta that did not change after longer BFA incubation times. The BFA treatment of cells co-expressing GFP-AtSYP31 and MNS3-mRFP showed a co-

localization of the fusion proteins on puncta (Supplementary Fig. 2a). In comparison, GFP-AtCASP highlighted a mixture of ER and punctate structures of varying size after 1 h BFA treatment (Fig. 4c). In contrast, the GFP-fused *cis*-Golgi membrane marker AtERD2, a recycling receptor for K/HDEL tetrapeptide-containing ER-resident proteins[42,43], was fully absorbed into the ER upon BFA treatment (Fig. 4d). Co-expression of MNS3-GFP with MNS1-mRFP showed that MNS1-mRFP, similar to AtERD2, was redistributed to the ER, whereas MNS3-GFP mainly remained on puncta as usually (Supplementary Fig. 2b). All in all, our data are in agreement with previous observations that distinct *cis*-Golgi membrane proteins behave very differently in the presence of BFA and that MNS3 belongs to the small group of *cis*-Golgi membrane proteins that after Golgi disassembly predominantly remain on membrane structures that have previously been termed Golgi remnants[38] or GECCO[40].

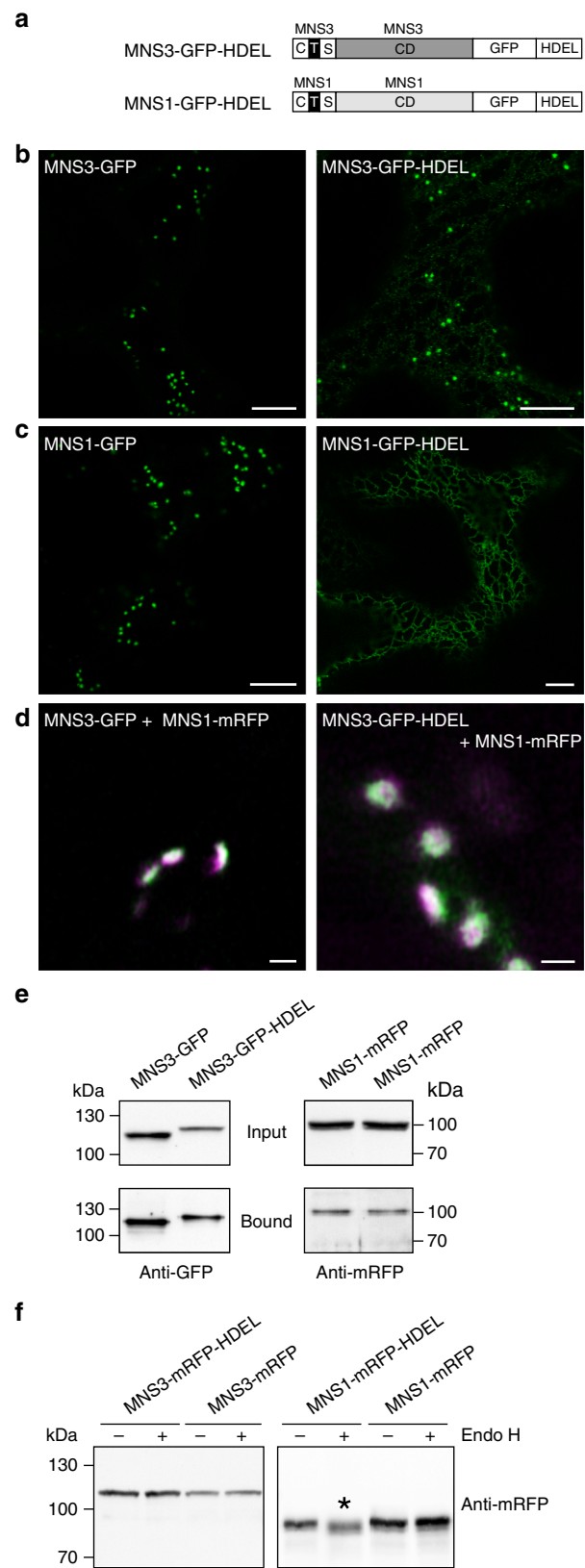

**Fig. 5** HDEL-mediated ER retrieval does not induce the redistribution of MNS3 to the ER. **a** Schematic illustration depicting the domain structure of the protein fusions MNS3-GFP-HDEL (full-length MNS3 C-terminally tagged with GFP and the ER retrieval signal HDEL) and MNS1-GFP-HDEL. **b**, **c** Confocal images (2 dpi) showing *N. benthamiana* leaf epidermal cells transiently expressing (**b**) MNS3-GFP or MNS3-GFP-HDEL, and (**c**) MNS1-GFP or MNS1-GFP-HDEL, respectively. Scale bars = 10 μm. **d** High-resolution Airyscan images of Golgi stacks from *N. tabacum* leaves labelled with MNS3-GFP (green) and MNS1-mRFP (magenta) or MNS3-GFP-HDEL (green) and MNS1-mRFP (magenta). Scale bar = 1 μm. **e** MNS3-GFP and MNS3-GFP-HDEL, respectively, were transiently co-expressed with MNS1-mRFP in *N. benthamiana* leaves. The GFP baits were purified using GFP-Trap beads and purified proteins were analyzed by SDS-PAGE and immunoblotting with anti-GFP and anti-mRFP antibodies. "Input" denotes total protein extracts before incubation with GFP-coupled beads; "bound" denotes the immunoprecipitated fraction. Source data are provided as a Source Data file. **f** Crude protein extracts from leaves of *N. benthamiana* ΔXF plants transiently expressing MNS3-mRFP-HDEL, MNS3-mRFP, MNS1-mRFP-HDEL, or MNS1-mRFP were subjected to Endo H digestion followed by immunoblotting with anti-mRFP antibodies. The asterisk marks the mobility shift. Source data are provided as a Source Data file

the luminal C-terminus of recombinant MNS3-GFP (MNS3-GFP-HDEL; Fig. 5a) assuming that this would facilitate the Golgi-to-ER retrieval of MNS3. A similar approach has recently been used to retrieve HDEL-tagged STtmd-YFP (STtmd fused to yellow fluorescent protein) from the Golgi to the ER[43]. Both the wild-type and the HDEL-tagged MNS3 fusion proteins were transiently expressed in *N. benthamiana* leaves and examined using confocal microscopy. As expected, wild-type MNS3-GFP exclusively labeled Golgi stacks (Fig. 5b). MNS3-GFP-HDEL, however, highlighted mainly punctate structures resembling Golgi stacks and very weakly the ER. Puncta co-localized with Golgi stacks either labeled with GnTI-mRFP or MNS1-mRFP (Supplementary Fig. 3a). To exclude the possibility that GFP-HDEL-mediated retrieval was not functional, we expressed a GFP-HDEL-tagged version of MNS1 as a control (Fig. 5a). In contrast to wild-type MNS1-GFP, MNS1-GFP-HDEL was completely shifted from the Golgi to the ER (Fig. 5c). High-resolution Airyscan imaging showed some overlap of the co-localizing MNS1-mRFP and MNS3-GFP-HDEL signals, which was comparable to the overlap of co-expressed MNS1-mRFP and MNS3-GFP (Fig. 5d). It was previously shown that *cis*/medial-Golgi resident N-glycan processing enzymes form heterodimers[24]. To examine whether GFP- and GFP-HDEL-tagged MNS3 interact with MNS1, we performed co-immunoprecipitation (co-IP) analysis. Immunoblotting revealed that both MNS3 variants interacted to a similar extent with MNS1 (Fig. 5e). A comparable interaction pattern was detected when the *cis*/medial-Golgi enzyme GnTI-mRFP was co-expressed (Supplementary Fig. 3b). Since an ER residency is typically marked by the presence of oligomannosidic N-glycans on glycoproteins, we hence performed digestions with Endo H. MNS3-mRFP-HDEL and MNS3-mRFP, as well as MNS1-mRFP-HDEL and MNS1-mRFP (MNS1 contains four predicted N-glycosylation sites, two of them with high probability), respectively, were transiently expressed in *N. benthamiana* ΔXF plants that almost completely lack complex N-glycans with core α1,3-fucose[10] and protein extracts were subjected to Endo H digestion. Immunoblotting with antibodies against mRFP revealed a small, but discernible mobility shift of the ER-located MNS1-mRFP-HDEL sample, whereas the other fusion proteins were insensitive (Fig. 5f). As only MNS1-mRFP-HDEL was Endo H sensitive, we concluded that MNS3-mRFP-HDEL, as well as MNS3-mRFP and MNS1-mRFP are in contact with

**Forced ER retrieval does not relocate MNS3 to the ER.** Blocking traffic between the ER and Golgi with BFA causes the redistribution of Golgi proteins to the ER in an invasive manner. To investigate the dynamic and very distinct behavior of MNS3 in a noninvasive way, we hence exploited the K/HDEL-mediated ER retrieval system. We appended the HDEL tetrapeptide signal to

Golgi-resident N-glycan processing enzymes. Collectively, these data show that with regards to its localization HDEL-tagged MNS3 essentially behaves like MNS3 indicating that the mechanism of HDEL-mediated ER retrieval is overruled by another, yet unknown mechanism that robustly retains MNS3 on punctate structures.

It has been shown for numerous Golgi-resident N-glycan processing enzymes that proper Golgi targeting and retention is dictated by their CTS regions and signals therein[11,25,44]. We hence tested whether the CTS region of MNS3 is responsible for retaining MNS3-GFP-HDEL on punctate structures. For this reason, we exchanged the MNS3-CTS region of the recombinant MNS3-GFP-HDEL fusion protein with that of MNS1 (and vice versa; Fig. 6a) and examined the subcellular localization of the resulting chimeric fusion proteins (MNS1-MNS3-GFP-HDEL vs. MNS3-MNS1-GFP-HDEL). Strikingly, the fusion protein containing the MNS3-CTS region (MNS3-MNS1-GFP-HDEL) was largely retained on punctate structures, whereas the MNS1-CTS expressing fusion protein (MNS1-MNS3-GFP-HDEL) was quantitatively shifted to the ER (Fig. 6b). In agreement with our confocal data are Endo H treatments and the immunoblotting of protein extracts of *N. benthamiana* ΔXF plants transiently expressing MNS1-MNS3-GFP-HDEL and MNS3-MNS1-GFP-HDEL, which revealed a mobility shift of the MNS1-MNS3-GFP-HDEL protein band indicating the presence of ER-typical oligomannosidic N-glycans (Fig. 6c), whereas MNS3-MNS1-

GFP-HDEL was Endo H insensitive. These findings show that the very specific MNS3 localization is determined by its CTS region.

**A cytoplasmic signal motif retains MNS3 in the Golgi.** Different classes of targeting signals have been identified in the cytoplasmic domains of transmembrane proteins from yeast, mammals, and plants[45–47]. Consequently, we speculated that the cytoplasmic tail of MNS3 carries specific signals that promote the concentration of MNS3 in the Golgi apparatus. To test the relative contribution of cytoplasmically exposed amino acid sequences to MNS3 localization, we used the construct expressing the MNS3-GFP-HDEL fusion protein to generate deletion constructs thereof that lacked different numbers of amino acids from the cytoplasmic tail (Fig. 7a) and observe an immediate effect on MNS3 localization. Remarkably, the tail deletion constructs expressing MNS3-88-GFP-HDEL (lacking amino acids at position 2–22), MNS3-98-GFP-HDEL (lacking aa 2–12), and MNS3-102-GFP-HDEL (lacking aa 2–8) were completely shifted to the ER, whereas MNS3-106-GFP-HDEL (lacking aa 2–4) was found predominantly in Golgi-like puncta (Fig. 7b). It seems that the very distinct behavior of MNS3 is linked to a sequence of amino acids (leucine, proline, tyrosine, serine; LPYS) within the truncated cytoplasmic tail of the MNS3-106-GFP-HDEL fusion protein that potentially confer Golgi retention or interfere with HDEL-mediated ER retrieval. To determine whether these amino acids are conserved among plants, we used the N-terminal 41 amino acids of the MNS3 cytoplasmic tail domain to retrieve and align similar plant protein sequences from the NCBI protein database, and then visualized the consensus sequence from the alignment using weblogo[48]. The first N-terminal 23 amino acids of the MNS3 tail, including the LPYS stretch, are highly conserved within the plant kingdom (Supplementary Fig. 4). Within the LPYS sequence, serine appears less conserved than the hydrophobic amino acids leucine and proline or the polar amino acid tyrosine. Based on this knowledge, we hence tested the effect of single amino acid substitutions on MNS3 localization by using the construct expressing the MNS3-106-GFP-HDEL fusion protein to replace leucine, proline or tyrosine with alanine (Fig. 7b). Live-cell imaging of the respective fusion proteins revealed that the MNS3-106Y-GFP-HDEL fusion (tyrosine replaced with alanine) highlighted mainly Golgi-like puncta and weakly the ER, whereas MNS3-106P-GFP-HDEL (proline replaced with alanine) predominantly labeled the ER (and less Golgi-like puncta). Remarkably, the MNS3-106L-GFP-HDEL fusion protein (leucine replaced with alanine) exclusively labelled the ER, indicating complete ER retrieval of the fusion protein. The exchange of leucine for alanine in the full-length cytoplasmic tail (MNS3-L5A-GFP-HDEL) similarly resulted in the relocation of MNS3 to the ER (Fig. 7b). This finding demonstrates that the hydrophobic leucine residue within the LPYS sequence is critical for the retention of MNS3 in the Golgi apparatus.

To test whether the removal of the critical LPYS sequence affects the steady-state localization of MNS3 in the Golgi, we expressed MNS3-GFP, MNS3-L5A-GFP, and MNS3-102-GFP without the HDEL tag under the control of the ubiquitin-10 gene promoter (UBQ10) in *N. benthamiana* (Fig. 8a). All three fusion proteins labeled Golgi stacks (Fig. 8b). Strikingly, both UBQ10::MNS3-L5A-GFP and UBQ10::MNS3-102-GFP were completely redistributed to the ER after 2 h BFA treatment, whereas UBQ10::MNS3-GFP remained on punctate structures. A similar behavior was observed in BFA-treated Arabidopsis lines stably expressing UBQ10::MNS3-GFP or UBQ10::MNS3-L5A-GFP (Supplementary Fig. 13) and BFA-treated *N. benthamiana* cells expressing cytoplasmic tail deletions and/or mutations of fusions of the MNS3-CTS region to GFP (Supplementary Fig. 5), thus revealing

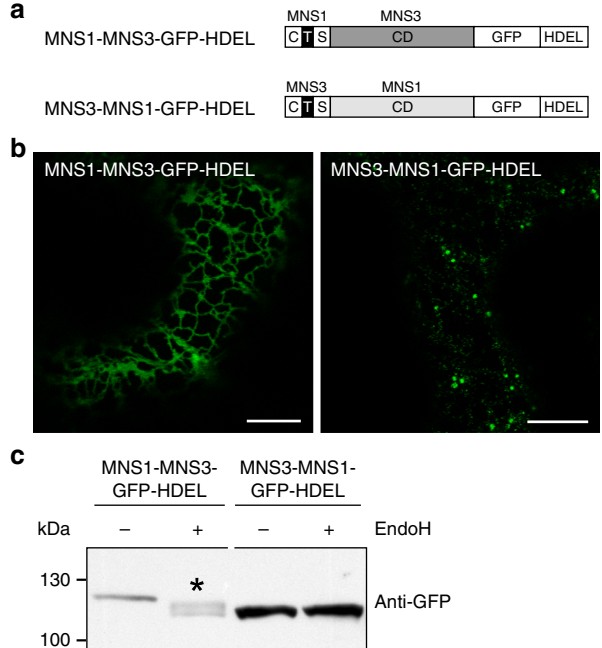

**Fig. 6** The MNS3-CTS region contains signals that overrule HDEL-mediated ER retrieval. **a** Schematic illustration depicting the domain structure of the protein fusions MNS1-MNS3-GFP-HDEL, containing the MNS1-CTS region and MNS3 catalytic domain (CD), and MNS3-MNS1-GFP-HDEL, containing the MNS3-CTS region and MNS1 catalytic domain, both C-terminally tagged with GFP and the ER retrieval signal HDEL. **b** Confocal images showing *N. benthamiana* leaf epidermal cells transiently expressing MNS1-MNS3-GFP-HDEL or MNS3-MNS1-GFP-HDEL. Scale bars = 10 μm. **c** Crude protein extracts from leaves of *N. benthamiana* ΔXF plants transiently expressing MNS1-MNS3-GFP-HDEL or MNS3-MNS1-GFP-HDEL were subjected to Endo H digestion followed by immunoblotting with anti-GFP antibodies. The asterisk marks the mobility shift. Source data are provided as a Source Data file

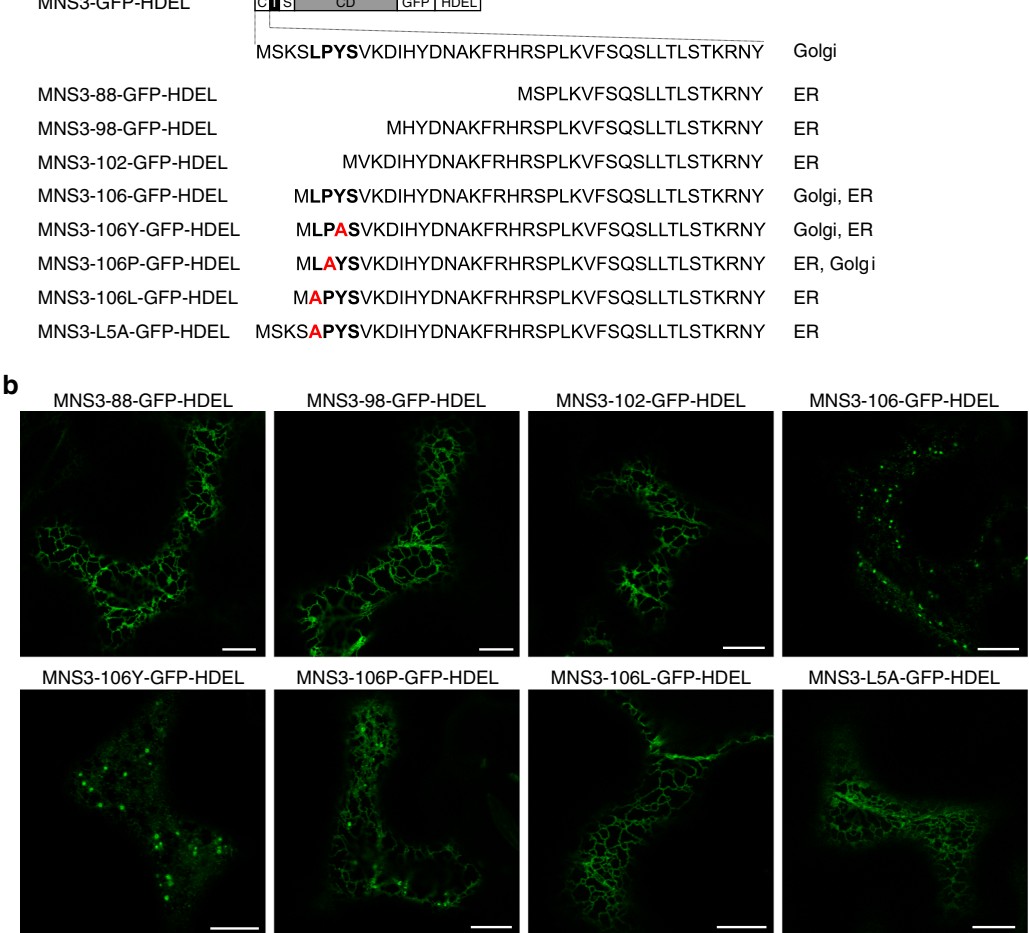

**Fig. 7** The cytoplasmic tail of MNS3 contains an amino acid sequence motif that promotes its Golgi retention. **a** The modular organization of MNS3-GFP-HDEL cytoplasmic tail deletion and mutation constructs and the subcellular localization of fusion proteins in *N. benthamiana* leaf epidermal cells are shown. The amino acid sequence of the cytoplasmic tail is given. **b** Confocal images of representative cells showing the subcellular localization of MNS3-GFP-HDEL tail-mutated fusion proteins. Images were acquired 2 dpi. Scale bars = 10 μm

that the cytoplasmically-exposed leucine most distal to the transmembrane domain is critical for the Golgi retention of MNS3. Notably, Endo H and PNGase F digestion of UBQ10::MNS3-L5A-GFP and UBQ10::MNS3-102-GFP showed that the majority of the protein carries Golgi-processed complex N-glycans with α1,3-fucosylation (Supplementary Fig. 11) indicating that the mutant MNS3 variants are competent for ER export and travel to the medial-to-*trans*-Golgi similar to MNS3.

**Predominant Golgi localization prevents MNS3-mediated ERAD.** To examine whether the specific localization of MNS3 is important to prevent the disordered degradation of misfolded glycoproteins in the ER, we targeted MNS3 to the ER. To this end, the MNS3-CTS region responsible for the specific MNS3 localization was replaced with the CTS region from Arabidopsis GCSI. GCSI catalyzes the first N-glycan processing step in the ER (Fig. 1a) and its CTS region is sufficient to retain attached proteins in the ER[11,25]. First, we expressed the chimeric GCSI-MNS3-GFP protein in *N. benthamiana* leaf epidermal cells to confirm the ER localization (Supplementary Fig. 6a). Next, GCSI-MNS3-GFP was expressed under the control of the MNS3 promoter or the UBQ10 promoter in Arabidopsis *bri1-5* plants that display a growth defect due to the ERAD of a misfolded BRI1 receptor (BRI1-5 protein[49]). The BRI1-5 protein has a conserved

cysteine changed to tyrosine, which leads to misfolding and its ER retention by different mechanisms followed by subsequent degradation. Notably, ERAD is regulated in plants and not all of the misfolded BRI1 receptor is sent for degradation in the *bri1-5* mutant. We reasoned that an enhanced trimming of mannose residues in *bri1-5* leads to a more severe growth defect, as more BRI1-5 protein will carry the processed glycan signal that is required for its degradation. Such an increased *bri1-5* phenotype has previously been described for *mns45 bri1-5* plants over-expressing MNS4 or MNS5 or in the *alg3 bri1-5* double mutant that constantly exposes a glycan degradation signal[50]. Whereas *bri1-5* expressing MNS3-mRFP displayed a *bri1-5* growth phenotype, *bri1-5* GCSI-MNS3-GFP plants were considerably smaller and showed a more severe morphological phenotype (Fig. 9a). Confocal microscopy revealed that GCSI-MNS3-GFP is located in the ER in *bri1-5* plants (Fig. 9b). Endo H digestion of UBQ10::GCSI-MNS3-GFP protein extracts from *N. benthamiana* wildtype plants led to a shift in mobility indicating the presence of ER-typical oligomannosidic N-glycans (Fig. 9c, Supplementary Fig. 6b). In contrast to UBQ10::GCSI-MNS3-GFP, UBQ10::GCSI-AtGnTI-GFP did not enhance the *bri1-5* phenotype indicating that the effect is not linked to the expression of the GCSI-CTS region (Supplementary Fig. 12). To further confirm this, we replaced the GCSI-CTS region of UBQ10::GCSI-MNS3-GFP with the one from the Arabidopsis cytokinin oxidase/dehydrogenase

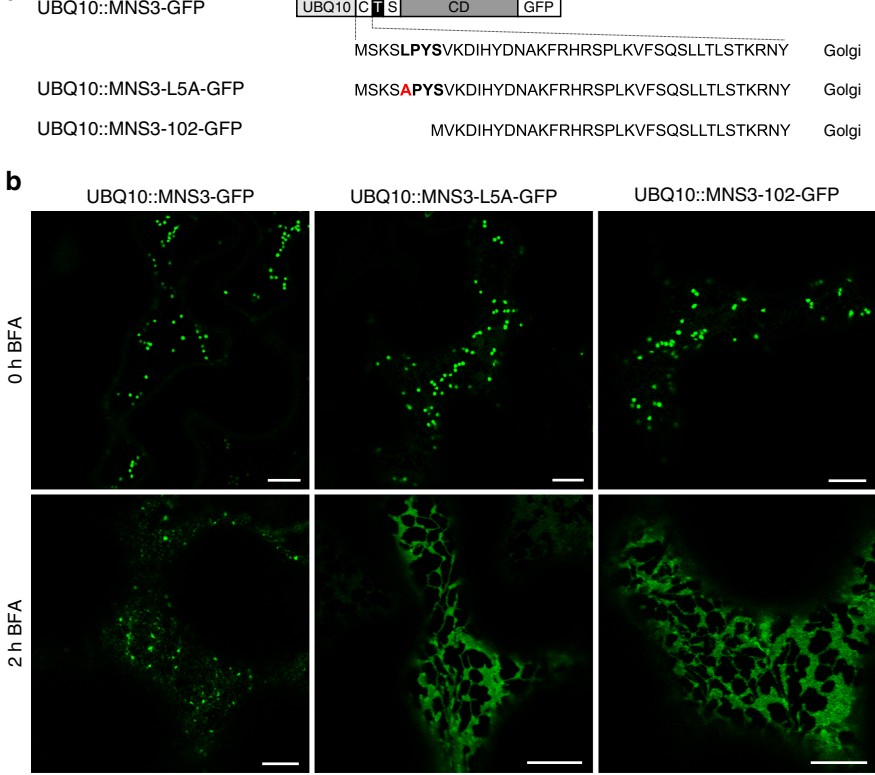

**Fig. 8** Subcellular localization of MNS3 tail mutants before and after BFA treatment. **a** Schematic illustration depicting the domain structure of MNS3-GFP carrying the wild type, mutated (L5A), or truncated (102) cytoplasmic tail expressed under the control of the UBQ10 promoter. **b** Confocal images of representative *N. benthamiana* leaf epidermal cells showing the subcellular localization of wild-type UBQ10::MNS3-GFP, mutated UBQ10::MNS3-L5A-GFP, and truncated UBQ10::MNS3-102-GFP before and after 2 h of BFA treatment. Images were acquired 2 dpi. Scale bars = 10 μm

CKX1, which is a type II membrane protein that predominantly localizes to the ER[51]. Consistent with our hypothesis, UBQ10::CKX1-MNS3-GFP expression enhanced the *bri1-5* phenotype (Supplementary Fig. 12).

Notably, *bri1-5* plants expressing MNS3::MNS3-L5A-GFP that contains the alanine-for-leucine substitution in the full-length cytoplasmic tail showed an enhanced *bri1-5* growth defect similar to that observed for GCSI-MNS3-GFP expressing *bri1-5* plants. *Bri1-5* expressing MNS3::MNS3-GFP again displayed the typical *bri1-5* growth phenotype (Fig. 9d). The MNS3-L5A effect on *bri1-5* growth was specific for *bri1-5* as backcrossing to wild type caused the loss of the phenotype (Supplementary Fig. 10). All in all, these data suggest that the retention of MNS3 in the ER has a severe effect on the growth phenotype of *bri1-5* due to the enhanced degradation of BRI1-5 by ERAD.

## Discussion

A localization of MNS3 in the Golgi apparatus is in agreement with several independent Arabidopsis proteomics studies that have demonstrated the presence of MNS3 in Golgi membrane fractions[52–54]. Such a localization is fully consistent with its enzymatic function in the regular N-glycan processing of properly folded secretory glycoproteins in the Golgi apparatus, where it generates the N-glycan substrate required for the subsequent processing by *cis*/medial-Golgi enzymes MNS1/MNS2 and proper complex-type N-glycan formation (Fig. 1a). High-resolution imaging of individual Golgi stacks revealed that co-expressed MNS3 and MNS1 largely co-localized (a marginally shifted overlap of both proteins was observed in Fig. 5d). This clearly suggested a localization of MNS3 in the vicinity of MNS1, which

is supported by FRET-combined fluorescence lifetime imaging microscopy (FLIM) and co-IP experiments that have shown that a portion of MNS3-GFP interacts with MNS1-mRFP, probably weakly, in plants and in vitro, respectively (Supplementary Fig. 7). Since FRET is strongly distance dependent, a residency of MNS3 in the vicinity, presumably upstream of MNS1 in *cis*-Golgi membranes is in full agreement with the assembly line model for N-glycan processing and the proposed three-stage model of Golgi structure and function[55].

The molecular determinants that establish and/or maintain the very specific MNS3 localization were greatly unknown up to now. We have previously shown that the CTS region contains all the information to concentrate MNS3 in the Golgi. This is in agreement with many other reports that the CTS region (particularly the cytoplasmic tail) plays a pivotal role in establishing and maintaining the steady-state localization of Golgi glycosyltransferase and glycosidases in yeast, mammals, and plants[45,56]. In contrast to yeast and mammals, the molecular determinants such as proteins that bind to the cytoplasmic regions of Golgi glycosylation enzymes or specifically recognize amino acid motifs therein are still elusive in plants[47,57,58].

The ER localization of the yeast MNS3 ortholog relies on the interaction of its transmembrane domain with Rer1p, a Golgi-localized retrieval receptor, which cycles the protein from the Golgi back to the ER[59]. In contrast, a mammalian study showed that the N-terminal cytoplasmic tail of human ERManI contains two di-basic arginine motifs (RRXX) that are required for binding to γCOP, the gamma subunit of COPI[21]. Although the MNS3 cytoplasmic tail contains numerous basic amino acids, which are essential residues of the binding motifs in cytoplasmic tails of Golgi glycosyltransferases and glycosidases, no recognizable

canonical COPI binding motif is present. The identified LPYS motif within the cytoplasmic tail of MNS3, however, appears unrelated to any of the so far described Golgi localization signals from yeast, mammals, and plants[46,60], despite its high conservation among plant ER-α-mannosidases (Supplementary Fig. 4).

To investigate the possibility that COPI subunits are involved in the Golgi localization of MNS3, we transiently co-expressed MNS3-GFP as well as UBQ10::MNS3-102-GFP (lacking the LPYS motif) with RNAi constructs that silence endogenous δCOP and εCOP in *N. benthamiana* leaves[29]. Interestingly, the Golgi localization of both the wildtype and the truncated MNS3 fusion proteins were partially shifted to the vacuole in the presence of δCOP or εCOP RNAi constructs (Supplementary Fig. 9). While this finding suggests an involvement of COPI-mediated recycling from *trans*- to *cis*-Golgi cisternae, it does not explain the observed differences between the MNS3 variants.

Our experiments convincingly show that the LPYS motif robustly retains MNS3 on Golgi membrane structures possibly similar to the ERGIC[61], *cis* initiators[62], GECCO[40], or Golgi remnants[38] after Golgi disassembly through BFA or SAR1-GTP co-expression. Interesting is the importance of leucine residues in several different ER-Golgi localization signals. Only recently, a conserved di-leucine motif (LXL) near the cytoplasmic C-terminus of the plant K/HDEL receptor AtERD2 was described to be essential for its Golgi residency and biological function in mediating ER retention of soluble ligands[43]. Replacement of leucine residues resulted in a significant shift of AtERD2 to the ER. Whether this effect was due to a defective ER export or accelerated Golgi-to-ER transport is still elusive. In the case of the LPYS motif, the Golgi-to-ER shift could equally point to a defective Golgi retention mechanism as a result of an abolished binding interaction between the cytoplasmic motif and so far unknown determinants in the Golgi apparatus. A preliminary proteomics approach to identify interacting proteins has yielded a set of ER- and Golgi-located candidate proteins with known (regulatory) functions in cargo selection and transport. An interaction with proteins such as AtSYP31 or AtCASP that co-localize with MNS3 in *cis*-Golgi membranes has not been observed so far.

Indicative of a defective Golgi retention is the observation that MNS3 lacking the LPYS motif is fully redistributed to the ER in response to BFA, whereas wild-type MNS3 was found in a dispersed punctate pattern. Such a behavior is rather exceptional since BFA normally shifts Golgi-resident glycosyltransferases and glycosidases back to the ER. Only a few Golgi proteins have so far been reported to show a similar response to BFA[35,41,63–68]. Notably, all of these proteins including MNS3 are integral membrane proteins that have been localized in *cis*-Golgi membranes and fulfil key functions at the plant ER-Golgi interface. We propose that BFA-induced punctate structures contain a minimal set of *cis*-Golgi membrane proteins with specific roles at the ER-Golgi interface (protein sorting, ER-Golgi tethering, and glycosylation reactions) and can serve as "seeds" to rapidly build up *cis*-Golgi cisternae by receiving materials directly from the ER[36,41,69]. The presence of MNS3 amidst these proteins suggests that it may be an important component of the not so well-characterized ER-Golgi interface in plants consisting of ERES, COPII/COPI vectors, and *cis*-most Golgi membranes. The specific MNS3 subcellular accumulation may ensure that ER residents and secreted glycoprotein cargo are efficiently trimmed from $Man_9GlcNAc_2$ to $Man_8GlcNAc_2$ structures. The MNS3-labeled compartment may also serve as a checkpoint for the separation of ERAD substrates and ER residents from secretory cargo that receive further processing in the Golgi.

Our results raise the question how such a steady-state localization can be reconciled with the biosynthetic and biological function of MNS3 typically performed in the ER. As mentioned earlier, MNS3 is dispensable for the ERAD of misfolded glycoproteins in plants[8,50]. Removal of a terminal mannose residue by MNS4 or MNS5 generates the α1,6-linked mannose on the C-branch, which constitutes the glycan degradation signal and diverts misfolded proteins to ERAD[8,70]. Here, we propose that MNS3 enhances the ERAD of misfolded glycoproteins like BRI1-5 when it is relocated to the ER. In vitro activity assays have shown that MNS3 preferentially cleaves off the terminal mannose residue from the B-branch of N-glycans and the *mns12* double mutant with an active MNS3 accumulates mainly $Man_8$-$GlcNAc_2$[7]. These findings indicate that MNS3 in its native subcellular environment (*cis*-Golgi) does not efficiently trim the mannose residue from the C-branch that would generate the N-glycan signal for degradation. However, when MNS3 is mis-targeted to the ER it promotes the generation of the glycan signal for degradation. ER-retained MNS3 could interact with other ER-resident ERAD factors like MNS4/MNS5 and modulate their activity toward enhanced degradation of misfolded proteins. Alternatively, MNS3 contributes directly to the generation of the exposed α1,6-linked mannose on the C-branch by enzymatic hydrolysis of the terminal α1,2-linked mannose. A prolonged contact time between ER-targeted MNS3 and ER-resident glycoproteins could facilitate the processing of additional mannose residues that normally are not processed when proteins are in contact with Golgi-located MNS3 (Fig. 9e).

Moreover, our results raise the question of how such a steady-state localization can be reconciled with its N-glycan processing function. Are ER-resident glycoprotein substrates transported through the *cis*-Golgi in order to receive processing by Golgi-resident MNS3 or does MNS3 cycle between the ER and the *cis*-most Golgi to fulfil its biosynthetic function? The finding that MNS3 interferes with ERAD of glycoproteins in the ER implies that it does not actively cycle between the ER and Golgi. Therefore, we propose that secretory, as well as ER-resident proteins cycle from the ER through the MNS3 compartment, where the mannose from the B-branch is removed (Fig. 9e). ERAD substrates likely also cycle through this compartment, because their N-glycans normally lack mannose residues on the B- and C-branches[8,50]. This could be a deliberate process to ensure or control N-glycan trimming for a yet unknown function (e.g., mannose trimming as a timer to control protein half-life of ER-residents) or is some kind of by-product of another process that requires recycling through the MNS3 compartment. Despite the observed interaction between MNS3 and MNS1, the typically low amounts of fully processed $Man_5GlcNAc_2$ on ER-resident glycoproteins suggest further that MNS3 and MNS1 activities are spatially separated from each other in the Golgi. Their interaction could therefore be transient during cycling of MNS3 through different Golgi compartments. Based on our findings and the predicted model for MNS3 localization and function we propose that the cytoplasmic domain constitutes a specific Golgi retention signal that is important for the spatial separation of MNS3 and ERAD substrates.

## Methods

**Plant material and growth conditions**. *A. thaliana* (ecotype Columbia-0) wild-type and mutant plants were grown in long-day conditions (16-h-light/8-h-dark photoperiod) on soil or on 0.5 × Murashige and Skoog (MS) medium containing 1% sucrose and 0.8% agar in a growth chamber or cabinet set to 22 °C. The mutants *mns3*[7], *fut11 fut12*[31], and *bri1-5*[50] were available from previous studies. *N. benthamiana* `wildtype` and ΔXF[10] plants, used for transient expression, were grown in a growth chamber at 24 °C with a 16-h-light/8-h-dark photoperiod for 5 weeks. *N. tabacum*, cultivar Petit Havana SR1, was grown throughout the year in a greenhouse at 21 °C with a 14-h-light/10-h-dark photoperiod, and a few days before/after infiltration was kept in a growth cabinet at 21 °C with a 12-h-light/12-h-dark photoperiod.

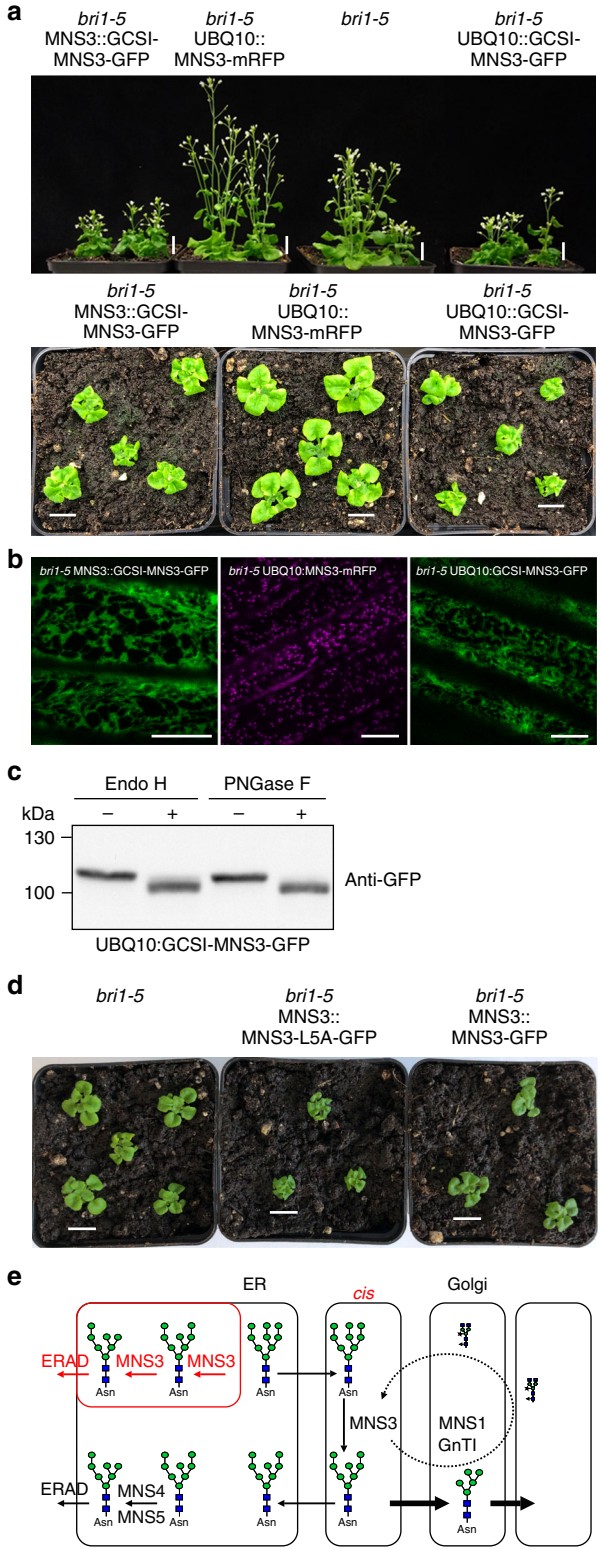

**Fig. 9** ER retention of MNS3 enhances the *bri1-5* phenotype. **a** Phenotype of Arabidopsis *bri1-5* and transgenic *bri1-5* plants expressing either MNS3::GCSI-MNS3-GFP, UBQ10::MNS3-mRFP, or UBQ10::GCSI-MNS3-GFP. Images of 5-week-old (upper panel) or 26-day-old (lower panel) soil-grown plants are shown. Scale bars = 1 cm. **b** Confocal images of transgenic *bri1-5* plants expressing either MNS3::GCSI-MNS3-GFP (green), UBQ10::MNS3-mRFP (magenta), or UBQ10::GCSI-MNS3-GFP (green). Scale bar = 15 μm. **c** Crude protein extracts from leaves of *N. benthamiana* wildtype plants transiently expressing UBQ10::GCSI-MNS3-GFP were subjected to Endo H and PNGase F digestion. Proteins were separated by SDS-PAGE and blots were probed with anti-GFP antibodies. Source data are provided as a Source Data file. **d** Phenotype of *bri1-5* and transgenic *bri1-5* plants expressing either MNS3::MNS3-L5A-GFP or MNS3::MNS3-GFP. Images of 22-day-old soil-grown plants are shown. Scale bars = 1 cm. **e** Proposed model for MNS3-catalyzed mannose trimming as part of regular N-glycan processing of glycoproteins in the Golgi or in the ER leading to enhanced ERAD. In the regular N-glycan processing pathway, ER-processed glycoproteins travel to the *cis*-Golgi, where they are modified by MNS3. Secretory proteins are further processed by Golgi-resident enzymes starting with MNS1 and GnTI, whereas ER-resident glycoproteins are cycled back to their home destination. MNS3-processed ERAD substrates are similarly returned to the ER, where they are subjected to demannosylation by MNS4/MNS5 generating the N-glycan signal triggering glycoprotein ERAD in plants. The presence of MNS3 in the ER promotes the generation of the glycan signal for degradation either by interacting with other ER-resident ERAD factors like MNS4/MNS5 and modulation of their activity toward enhanced degradation of misfolded proteins or by contributing directly to the generation of the exposed α1,6-linked mannose on the C-branch

similar to pPT2[74], but instead contains spectinomycin and hygromycin resistance genes for selection in bacteria and transgenic plants, respectively. The construct expressing MNS3-GFP-HDEL was generated by PCR amplification of the full-length MNS3 and GFP coding regions from p20-MNS3 using the primers At1g30000_12F/GFP_12R. The PCR product was XbaI/BglII-digested and cloned into the XbaI/BamHI-site of p62 (p62-MNS3). MNS1-GFP-HDEL was generated by PCR amplification of the full-length MNS1 and GFP coding regions from p20-MNS1 using the primers At1g51590_33F/GFP_12R. The PCR product was SpeI-digested and cloned into the XbaI-site of p62 (p62-MNS1). For the expression of MNS3-MNS1-GFP-HDEL, the MNS3-CTS region, MNS1 catalytic domain and GFP coding sequence were PCR amplified from the plasmid p87-MNS3 with primers At1g30000_40F/GFP_12R, digested with SpeI/BglII and cloned into the XbaI/BamHI-site of p62 (p62-MNS3$_{CTS}$-MNS1$_{CD}$). p87 contains a custom-made sequence of the MNS1 catalytic domain (Genelife) that was BglII/XbaI-digested and cloned into the XbaI/BamHI-site of vector p47, and carries the *A. thaliana* MNS1 promoter region that was PCR amplified from Col-0 genomic DNA using primers At1g51590_27F/At1g51590_28R and inserted into the KpnI/XbaI-site of p47. To generate MNS1-MNS3-GFP-HDEL, the DNA fragment containing the MNS1-CTS region, MNS3 catalytic domain, and GFP coding sequence was PCR amplified from the plasmid p77-MNS1 with primers At1g51590_41F/GFP_12R, XbaI/BglII-digested, and cloned into the XbaI/BamHI-site of p62 (p62-MNS1$_{CTS}$-MNS3$_{CD}$). For the generation of p62 constructs expressing GFP-HDEL tagged full-length MNS3 with various N-terminally truncated and/or mutated cytoplasmic tails, the respective CTS sequences were PCR amplified from p62-MNS3$_{CTS}$-MNS1$_{CD}$ with the following primer combinations: MNS3-106-GFP-HDEL was amplified with At1g30000_43F/38R, MNS3-102-GFP-HDEL was amplified with At1g30000_42F/38R, MNS3-98-GFP-HDEL was amplified with At1g30000_41F/38R, MNS3-88-GFP-HDEL was amplified with At1g30000_37F/38R, MNS3-106L-GFP-HDEL with At1g30000_46F/38R, MNS3-106P-GFP-HDEL was amplified with At1g30000_44F/38R, MNS3-106Y-GFP-HDEL was amplified with At1g30000_45F/38R, and MNS3-L5A-GFP-HDEL was amplified with At1g30000_47F/38R. All PCR products were XbaI/BamHI digested and inserted into p62-MNS1$_{CTS}$-MNS3$_{CD}$ by removing the MNS1-CTS region via XbaI/BamHI digestion. For the expression of MNS3-mRFP-HDEL or MNS1-mRFP-HDEL, the respective full-length MNS3 or MNS1 sequences were XbaI/BamHI- or SpeI/BamHI-digested and cloned into the XbaI/BamHI-site of p59, respectively (p59-MNS3 or p59-MNS1). p59 is similar to p31, but instead contains HDEL-tagged mRFP (mRFP-HDEL). To generate the constructs expressing MNS3-CTS102-GFP and MNS3-CTS106L-GFP, the respective CTS sequences were removed from p62-MNS3$_{CTS102}$-MNS3$_{CD}$ and p62-MNS3$_{CTS106L}$-MNS3$_{CD}$, respectively, by XbaI/BamHI digestion and ligated into the XbaI/BamHI site of the plant expression vector p20F[7]. For MNS3$_{CTSL5A}$-GFP expression, the XbaI/BamHI-digested MNS3-

**Plasmid construction and generation of transgenic plants**. The following constructs have been published previously: MNS3-GFP (p20-MNS3[7]), MNS3-CTS-GFP (p20-MNS3-CTS[7]), MNS1-GFP (p20-MNS1[24]), MNS1-mRFP (p31-MNS1[24]), STtmd-mRFP[27], GnTI-mRFP (p31-GnTI[11]), GFP-AtCASP[71], mRFP-AtCASP[72], AtERD2-GFP[73], and SAR1-GTP[12]. Unless stated otherwise, fusion proteins were expressed under the control of the cauliflower mosaic virus 35S promoter. To generate the vector p62 for expressing GFP-HDEL tagged fusion proteins, the expression cassette of the plant expression vector p61 that contains a custom-made 3×HA-HDEL sequence (GeneArt-Gene Synthesis) was removed by HindIII/EcoRI-digestion and cloned into the HindIII/EcoRI-site of vector p27. This vector is

CTSL5A sequence was cloned into the XbaI/BamHI-site of p20F (p20-MNS3$_{CTSL5A}$). To generate the vector expressing MNS3::MNS3-GFP, the 873 bp promoter region of *A. thaliana* MNS3[7] was amplified by PCR from Col-0 genomic DNA using primers At1g30000_30F/32R. The MNS3-CTS region and MNS3 catalytic domain were amplified from p20-MNS3 with primers At1g30000_33F/35R and At1g30000_34F/31R, respectively. These DNA fragments were assembled using the Gibson Assembly Cloning Kit (New England Biolabs) and subcloned into the StrataClone blunt PCR cloning vector pSC-B. The assembled DNA fragment was removed by KpnI/BglII-digestion and cloned into the KpnI/BamHI site of the plant expression vector p47[8] to create the vector construct p77-MNS3. The vector p47 is similar to p20F, but instead contains a UBQ10 promoter and a hygromycin resistance gene for selection of transgenic plants. For the expression of MNS3::MNS3-L5A-GFP, the MNS3-CTS region from vector p77-MNS3 was removed by XbaI/BamHI digestion and replaced with the XbaI/BamHI-digested MNS3-CTSL5A sequence (p77-MNS3$_{CTSL5A}$). To generate the vector expressing MNS3::GCSI-MNS3-GFP, the MNS3-CTS region from construct p77-MNS3 was removed by XbaI/BamHI digestion and replaced with the XbaI/BamHI-digested CTS region of *A. thaliana* GCSI[11] (p77-GCSI$_{CTS}$-MNS3$_{CD}$). For UBQ10::GCSI-MNS3-GFP expression, the MNS3 promoter region of p77-GCSI$_{CTS}$-MNS3$_{CD}$ was removed via KpnI/XbaI digestion and replaced with the KpnI/XbaI-digested UBQ10 promoter region of vector p47 (p47-GCSI$_{CTS}$-MNS3$_{CD}$). The construct expressing UBQ10::MNS3-GFP was generated by removing the GCSI-CTS region from construct p47-GCSI$_{CTS}$-MNS3$_{CD}$ via XbaI/BamHI digestion and replacing it with the XbaI/BamHI-digested MNS3-CTS region (p47-MNS3). The constructs expressing UBQ10::MNS3-L5A-GFP and UBQ10::MNS3-102-GFP were generated by removal of the GCSI-CTS region from the p47-GCSI$_{CTS}$-MNS3$_{CD}$ construct via XbaI/BamHI digestion and replacement with the XbaI/BamHI-digested MNS3-CTSL5A sequence (p47-MNS3$_{CTSL5A}$-MNS3$_{CD}$) and MNS3-CTS102 sequence (p47-MNS3$_{CTS102}$-MNS3$_{CD}$), respectively. The p48-MNS3 construct expressing UBQ10::MNS3-mRFP was generated by cloning the MNS3-CTS region and catalytic domain via XbaI/BamHI digestion into vector p48 (like p47, but contains the mRFP coding sequence). To generate the construct GFP-AtSYP31, the SYP31 coding sequence (At5g05760) lacking the start codon was PCR amplified from Arabidopsis (Col-0) leaf cDNA using primers SYP31_3F/4R, XbaI-digested and inserted into the XbaI-linearized p37 vector (p37-AtSYP31). The vector p37 is similar to vector p20F, but instead is suited for N-terminal GFP tagging of the protein of interest. The vector expressing UBQ10::GCSI-AtGnTI-GFP was generated by PCR amplification of the GCSI-CTS region and Arabidopsis GnTI catalytic domain from vector p57-GCSI using the primer combination GCSI_7F and AthGnTI_9R, XbaI/BglII digestion of the PCR fragment and ligation into XbaI/BamHI-digested p47, generating p47-GCSI$_{CTS}$-AtGnTI$_{CD}$. The vector expressing UBQ10::CKX1-MNS3-GFP was generated by PCR amplification of the CKX1-CTS region from Arabidopsis (Col-0) leaf cDNA using primers CKX1_5F/6R, XbaI/BamHI digestion of the PCR fragment, and ligation into XbaI/BamHI-digested p47-GCSI$_{CTS}$-MNS3$_{CD}$, generating p47-CKX1$_{CTS}$-MNS3$_{CD}$. Primer sequences are listed in Supplementary Table 1. For stable expression in the *bri1-5* or *mns3* mutants, *A. thaliana* plants were floral dipped with *Agrobacterium tumefaciens* strain UIA143[74] containing the respective plasmid construct, and transgenic plants were later selected on MS plates containing appropriate antibiotics.

**Expression and subcellular localization of protein fusions.** Transient expression of fluorescent protein fusions in *N. benthamiana* or *N. tabacum* leaf epidermal cells was performed using the *A. tumefaciens* (strains UIA143 or GV3101)-mediated infiltration technique[23]. Briefly, liquid cultures of transformed agrobacteria were pelleted by centrifugation at 5000 rpm at room temperature for 5 min. The bacterial pellet was washed once and then resuspended with 1 ml infiltration medium (50 mM MES, 2 mM Na$_3$PO$_4$.12H$_2$O, 5 mg ml$^{-1}$ D-glucose, and 0.1 mM acetosyringone). The bacterial suspension was diluted with infiltration buffer to an optical density at 600 nm (OD$_{600}$) of 0.10 for all full-length MNS3 and MNS1 fusion proteins and fusions thereof; 0.05 for all MNS3-CTS fusion proteins, STtmd-mRFP, and mRFP-HDEL; 0.15 for GFP-AtCASP, mRFP-AtCASP, δCOP RNAi, and εCOP RNAi; 0.03 for SAR1-GTP, 0.10 for GFP-AtSYP31, AtERD2-GFP, UBQ10:GCSI-AtGnTI-GFP, and GnTI-mRFP; and 0.08 for DAD1-mRFP. For enzymatic deglycosylation and co-IP experiments, all fusion proteins were infiltrated at OD$_{600}$ 0.20. The final dilution was injected through the stomata on the abaxial leaf side using a 1-ml syringe. Infiltrated plants were kept in growth chambers prior to observation. For confocal microscopy, small segments of infiltrated leaf tissue (~3 × 3 mm) expressing the protein fusion(s) of interest were analyzed 2–4 days after infiltration. For FRET–FLIM experiments and co-localization analyses, leaf segments were treated with the actin-depolymerizing agent latrunculin B (Calbiochem, stock solution at 1 mM in DMSO) at a concentration of 25 µM for 45–60 min prior to image acquisition to inhibit Golgi movement[11,75]. BFA (Sigma, stock solution at 10 mg ml$^{-1}$ in DMSO) was used at a final concentration of 100 µg ml$^{-1}$, unless stated otherwise. Confocal images were obtained on confocal laser scanning microscopes from Zeiss (LSM 510, LSM 880 with Airyscan) or Leica (TCS SP5 with HyD, TCS SP8-gSTED with HyD). On the Zeiss LSM 510, samples were excited using 488- and 543-nm laser lines for GFP and RFP, respectively, and observed in multitrack mode with line switching using a 63 ×/1.4 NA DIC or 100 ×/1.4 NA oil immersion objective. Signals were collected using 505/530- and 585/615-nm band-pass filters for GFP and RFP, respectively.

For high-resolution imaging, images were acquired on a Zeiss LSM 880 using a 100 ×/1.46 NA DIC M27 Elyra oil immersion objective. Samples were excited using 488- and 561-nm laser lines for GFP and RFP, respectively, in multitrack mode with line switching. Signals were collected using the Airyscan detector with emission wavelengths of 523 nm for GFP and 579 nm for RFP. On the Leica SP5 or SP8, samples were excited using 488- and 561-nm laser lines for GFP and RFP, respectively, and observed using a 63 ×/1.4 NA or 100 ×/1.4 NA CS2 STED white oil immersion objective, respectively. Signals were collected simultaneously from 500 to 530 nm for GFP and 600 to 630 nm for RFP. Typically, 1024 × 1024 images were collected in 8-bit with 4- to 8-times line averaging and 1 Airy unit in xyz scan mode. Post-acquisition image processing was performed in Adobe Photoshop CS6.

**Co-localization analyses of co-expressed protein fusions.** Images of latrunculin B-treated cells expressing MNS3-GFP together with the *cis*/medial-Golgi protein MNS1-mRFP or the non-plant medial/*trans*-Golgi marker STtmd-mRFP were acquired 2 dpi under non-saturating conditions using a 512 × 512 image format, zoom factor 3, and a ×100/1.40 NA oil immersion objective of an inverted Zeiss LSM 510. The pinhole was set to 1 Airy unit, and background noise was reduced by 4–8 times line averaging. Only cells with comparable GFP and mRFP fluorescence levels were considered for analysis. To determine the MNS3/MNS1 and MNS3/STtmd co-localization ratios, 12 and 9 confocal images from two independent plants, respectively, were obtained and analyzed with the IMAGEJ 1.46 m plug-in JACOP[76] using the Pearson's correlation coefficient (*r*). Statistical analysis was performed in Microsoft Excel using a two-tailed Student's *t*-test for the comparison of two samples, assuming equal variances (degrees of freedom = 19; critical *t*-value = 2.1).Mean values and standard deviations of Pearson's *r* are given in the main text. The underlying source data are provided in the Source Data file.

**Antibodies.** The following primary antibodies were used in this study: mouse anti-GFP antibody (11814460001, Roche; 1:2000 dilution); mouse anti-RFP antibody (6G6, Chromotek; 1:2000 dilution); and mouse anti-alpha-tubulin antibody (T6074, Sigma; 1:5000 dilution). The secondary antibodies were as follows: rabbit anti-mouse IgG (whole molecule)–Peroxidase antibody (A9044, Sigma; 1:10,000 dilution); and goat anti-rabbit IgG (whole molecule)–Peroxidase antibody (A0545, Sigma; 1:100,000 dilution). For the generation of MNS3-specific antibodies, a peptide that corresponds to the MNS3 C-terminus was synthesized (LDKFVFNTEAHPLPIRRNT). The MNS3 peptide was coupled to keyhole limpet hemocyanin and then injected into rabbits. The peptides and antisera were prepared by Gramsch Laboratories (Schwabhausen, Germany). Affinity-purified antibodies were prepared using the SulfoLink Immobilization Kit for Peptides (44999, Thermo Scientific). Coupling of the peptides and subsequent affinity purification were performed following the instructions of the manufacturer. The antibody was used at a 1:2000 dilution. Its specificity was tested by immunoblotting studies with recombinant MNS3 produced as described[7] and protein extracts of Col-0 and *mns1 mns2 mns3*[7] knockouts (Supplementary Fig. 15).

**Co-immunoprecipitation assays using GFP-Trap®-A.** Transient expression of fluorescent protein fusions in *N. benthamiana* was performed by infiltration of leaves as described[11]. For co-expression experiments, resuspended agrobacteria were diluted to an OD$_{600}$ of 0.2 for MNS3-GFP, MNS3-GFP-HDEL, MNS1-mRFP, and GnTI-mRFP, respectively. Two hundred and fifty milligrams of infiltrated leaf material was harvested, ground in liquid nitrogen, and resuspended in 500 µl of extraction buffer containing 1 × PBS, 1% Triton X-100, and 1% protease inhibitor cocktail (Sigma). The tubes were placed on ice for 15 min with occasional mixing. The samples were centrifuged at 6000 rpm for 6 min at 4 °C and the resulting pellet was discarded. The clear supernatant was diluted 1:1 with dilution buffer containing 1 × PBS and 1% protease inhibitor cocktail. For co-IP assays, 15 µl GFP-Trap®-A beads (Chromotek) were washed twice with ice-cold dilution buffer and spun down at 4000 rpm for 4 min at 4 °C. The beads were resuspended with ice-cold dilution buffer and mixed with protein extracts followed by an incubation step with end-over-end mixing for 60–90 min at 4 °C. The samples were pelleted at 4000 rpm for 4 min at 4 °C. The beads were transferred onto Micro Bio-Spin chromatography columns (BioRad) and washed twice with 600 µl 1 × PBS. Beads were resuspended with 60 µl 2 × Laemmli sample buffer and boiled for 4 min at 95 °C. The dissociated immunocomplexes (referred as bound) were eluted by centrifugation and subjected to SDS-PAGE and immunoblot analysis using anti-GFP and anti-mRFP antibodies. The specificity of GFP-Trap®-A (and mRFP-Trap®-A) beads was validated as shown in Supplementary Fig. 14. Source data are provided as a Source Data file.

**Enzymatic deglycosylation.** Leaves of Arabidopsis or agroinfiltrated *N. benthamiana* plants were harvested, ground in liquid nitrogen, and resuspended in 1.5 × Laemmli sample buffer. Samples were boiled for 5 min at 95 °C and centrifuged until the supernatant was clear. The supernatant proteins were denatured and incubated with or without Endo H or PNGase F (both from NEB) according to the manufacturer's manual. Deglycosylated proteins and controls were then subjected to SDS-PAGE followed by immunoblot analysis with antibodies against endogenous MNS3 (anti-MNS3), GFP or mRFP.

**Glycan analysis**. Preparation of total N-glycans[31] was performed from 500 mg of plant material that was ground and resuspended in 2.5 ml of 5% formic acid and 0.1 mg ml$^{-1}$ pepsin. The slurry was incubated at 37 °C for 20 h with occasional stirring. Insoluble material was then removed by centrifugation. From the supernatant, glycopeptides were enriched by cation exchange and gel filtration[77]. Subsequently N-glycans were released from glycopeptides with peptide N-glycosidase A (Roche) and purified by cation exchange chromatography, gel filtration, and passage through a reversed phase matrix.[31] MALDI mass spectra were acquired using an Autoflex Speed mass spectrometer (Bruker). For the analysis of glycopeptides, N. benthamiana wild-type leaves were infiltrated with MNS3-GFP and purified using the GFP-Trap® kit (Chromotek) as described above[7,24]. Purified protein was subjected to SDS-PAGE under reducing conditions and Coomassie Brilliant Blue staining was performed to detect polypeptides. The corresponding band was excised from the gel, destained, carbamidomethylated, in-gel trypsin digested, and analyzed by LC–ESI–MS as described[78].

**FRET–FLIM data acquisition and analysis**. Samples were excised from infiltrated N. tabacum leaves and treated with latrunculin B for 1 h. 2P-FRET-FLIM data[24,79] capture was performed using a two-photon excitation microscope at the Central Laser Facility of the Rutherford Appleton Laboratory. Briefly, a two-photon microscope was constructed around a Nikon TE2000-U inverted microscope using custom-made XY galvanometers (GSI Lumonics) for the scanning system. Laser light at a wavelength of 920 ± 5 nm was obtained from a mode-locked titanium sapphire laser (Mira, Coherent Lasers), producing 180-fs pulses at 75 MHz, pumped by a solid-state continuous wave 532-nm laser (Verdi V18, Coherent Lasers). Two-photon excitation at 920 nm was chosen to allow reduced autofluorescence emission from chloroplast and guard cells. The laser beam was focused to a diffraction-limited spot through a Nikon VC × 60/1.2 NA water immersion objective and specimens illuminated at the microscope stage. Fluorescence emission was collected without descanning, bypassing the scanning system, and passed through a BG39 (Comar) filter to block the near infrared laser light. Line, frame, and pixel clock signals were generated and synchronized with an external fast microchannel plate photomultiplier tube (MCP-PMT, Hamamatsu R3809U) used as the detector. These were linked via a time-correlated single-photon-counting PC module SPC830 (Becker and Hickl) to generate the raw FLIM data. Prior to FLIM data collection, the GFP and mRFP expression levels in the plant specimens within the region of interest were confirmed using a Nikon eC1 confocal microscope with excitation at 488 and 543 nm, respectively. A 633-nm interference filter was used to minimize further the contaminating effect of chlorophyll autofluorescence emission that would otherwise obscure the mRFP emission. FLIM images were analyzed by obtaining excited-state lifetime values of a single cell and calculations together with image processing were made using the SPC Image analysis software (Becker and Hickl). Lifetime values were collected on a single pixel basis from the center of individual Golgi bodies and lifetimes were recorded in Microsoft Excel. Decay curves of a single point highlight an optimal single exponential fit when chi square ($\chi^2$) values are 1 (points with $\chi^2$ from 0.9 to 1.3 were taken). The collected data values were used to generate histograms depicting the distribution of lifetime values of all data points within the samples. Results are from two independent experiments; each experiment included two biological replicates (plants). In total, 15 cells (283 Golgi bodies) were analyzed for MNS3-GFP; 17 cells (269 Golgi bodies) for the MNS3-GFP/MNS1-mRFP combination, and 18 cells (301 Golgi bodies) for the MNS3-GFP/GnTI-mRFP combination. An observed protein–protein interaction is described by the decrease of the donor fluorescence lifetime (quenching) due to energy transfer to the acceptor, which can be calculated by measuring the fluorescence lifetime of the donor in the presence and absence of the acceptor and can be expressed as a percentage of the donor lifetime, a value referred to as "energy transfer efficiency" (E). The percentage efficiency (E%) can be calculated using Eq. (1)

$$E = \left[ 1 - \left( \frac{\tau_{DA}}{\tau_D} \right) \right] \times 100 \qquad (1)$$

where $\tau_{DA}$ and $\tau_D$ are the mean pixel-by-pixel excited-state lifetimes of the donor in the presence and absence of the acceptor determined for each pixel. Quenching of average donor lifetimes by a minimum of 0.2 ns or 8% in the presence of the acceptor was considered relevant to indicate protein–protein interaction. Since the instrument response (IR) in our setup is determined to be <60 ps, there was no need to deconvolute the IR function from the sample data decay curves. Thus, lifetime differences of larger than 100 ps can be easily resolved.

The most representative images are shown throughout the article. The experiments have been performed in triplicates or more unless stated otherwise.

**Reporting summary**. Further information on research design is available in the Nature Research Reporting Summary linked to this article.

## Data availability

The authors declare that all relevant data generated or analyzed during this study are included in this article and its supplementary information file. The source data underlying Figs. 2c, 5e, f, 6c, and 9c, Supplementary Figs. 3b, 4, 6b, 7, 10b, 11, 14, and 15, and co-localization analyses are provided in the Source Data file. All other data that support the presented findings are available from the corresponding author upon reasonable request.

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

## Acknowledgements

The work in this article was supported by the Austrian Science Fund (FWF): J2981-B20 and T655-B20 (to J.S.), P28218-B22 (to R.S.), an Oxford Brookes University fellowship (to V.K.), and Oxford Brookes University (to C.H.). We thank the STFC for funding access to the Central Laser Facility (grant no. 12130006 to C.H.). We thank Josephine Grass and Karin Polacsek (BOKU, Department of Chemistry) for help with MS analyses and the BOKU-VIBT Imaging Center for access and expertise.

## Author contributions

J.S., J.K., U.V., C.V., E.L., V.K. and R.S. performed research. J.S., J.K., U.V., C.V., S.W.B., F.A., C.H. and R.S. analyzed data. J.S. and R.S. designed the research and wrote the paper.

## Additional information

**Competing interests:** The authors declare no competing interests.

