## [Peer Review File · Nature Communications]

Reviewers' comments:

Reviewer #1 (Remarks to the Author):

In the submitted paper, Schoberer and collaborators studied the Golgi retention mechanism of the Arabidopsis ER alpha-mannosidase I (MNS3). Using different constructs containing the MNS3 protein fused with fluorescent reporter and drugs provoking the relocation of Golgi enzymes, they reexamine the subcellular localization of the full-length MNS3 and demonstrate that MNS3 resides exclusively in the Cis/medial-Golgi cisternae and that the specific Golgi retention is due to the presence of a tetrapeptide signal in the cytoplasmic tail of the Arabidopsis MNS3. While the work is well done and the paper written in a comprehensive manner, I feel that the results remain interesting for a subset of scientists which are in the plant glycobiology community.

Did the authors try to express a shorter tagged version of the MNS3 (e.g. c-myc; HA) in parallel to the fusion to the GFP? It would be worth proving that the behavior that they observed for the MNS3-GFP is not a special one due to the length of the MNS3-GFP and remains with a shorter tagged version. Did you try to express the MNS3-GFP in a stable manner in BY2 cells for example and observed the same behavior? Did they express the MNS3-HA-HDEL to confirm their results? Indeed, previous studies have shown that fusion with GFP induced abnormal subcellular localization and behavior of proteins.

When they analyzed their images and mentioned respectively 9 and 12 images, is these images came from 9 and 12 independent plants? It would be good for the reader to clarify how many independent plants have been studied for the calculation of the Pearson's correlation coefficient.

When the authors performed the experiments with the BFA treatment, did they try to time course and different BFA concentrations? Dis they observe the same behavior for the MNS3-GFP? How do they make sure that the MNS3 does not need more time to relocate in the ER?

Reviewer #2 (Remarks to the Author):

In this manuscript, Schoberer et al. first re-visited the cis-Golgi sub-cellular localisation of steady-state MNS3 via confocal microscopy through transiently over-expression in *N. benthamiana* leaves and native-promoter-driven expression in *A. thaliana* mutants. Then the authors further defined the cis/medial-Golgi localisation of MNS3, and demonstrated its N-glycosylation in medial/trans-Golgi. Schoberer et al. proved that unlike the majority of Golgi-localised proteins, MNS3 is insensitive to BFA treatment or to the GTP-locked form of SAR1 (SAR1-GTP), revealing its specific localisation in Golgi remnants possibly originating from incomplete disassembly of Golgi stacks upon the disruption of early secretory pathway. Golgin AtCASP and cis-Golgi protein SYP31 were proved to display similar pattern. In addition, the authors used HDEL-tagged MNS3 (MNS3-GFP-HDEL) and its mutation forms and characterised that the fifth leucine residue (L5) in the MNS3 cytoplasmic tail is essential for MNS3 Golgi localisation. The role of MNS3's right location for ERAD degradation of BRI is also studied here in vivo.

In summary, this manuscript provides novel evidences of the crucial amino acid L5 form LPYS motif in MNS3 cytoplasmic tail for determining its specific cis-Golgi localisation, and the importance of this localisation in ERAD degradation of BRI. However, this manuscript offered little information on the underlying mechanisms of how the L5 possibly affect the MNS3 localisation. Two hypothesis exist for the important function of the leucine residue or the LPYS motif in MNS3: 1. Does it affect MNS3 ER export? and 2. Does it facilitate MNS3 ER retrieval from cis-Golgi? If the answer is yes, further experiments could address the underlying mechanisms and thus the novelty of this study.

Major comments:

1. How could the authors tell the puncta shown in Figure 2B is Golgi apparatus without co-localizing with the known Golgi markers? To assure the Golgi localization of the MNS3-GFP under native promoter, crossing or labeling with known Golgi markers are needed. (Figure 2B)

2. It is wired that when co-expressing of SAR1-GTP with MNS3-GFP and ST-mRFP, only ST-mRFP will be trapped in the ER but MNS3 remains puncta localization. Since MNS3 is synthesized in the ER and delivered from the ER to the Golgi apparatus via the COPII vesicles, why blocking the COPII-mediated ER export will not result in any ER pattern of MNS3? (Figure 3B)

3. When treated with BFA for 150 min, the authors claimed that MNS3 exhibited weak ER pattern

(Figure 3C). Nevertheless, it is hard to tell whether this is ER pattern or just some small puncta. ER markers should be used to confirm it. (Figure 3C)

4. For the Co-IP assay, Free-GFP should be used as a negative control. (Figure 5E)

5. For the Endo H digestion, no significant shift of MNS1-mRFP-HDEL was observed! Lower percentage of SDS-PAGE should be used to differentiate the shift. (Figure 5F)

6. Why the MNS3-mRFP showed two bands in Figure 5F? (Figure 5F)

7. Deletion and mutation analysis showed that the N-terminal cytoplasmic tail of MNS3 contains a signal motif LPYS affecting the protein Golgi localization. Nevertheless, will the LPYS motif mutation affect normal ER export or proper protein folding of MNS3, resulting in the ER pattern? The authors could use the same assay in Figure 2 (PNGase F assay) to see whether the mutants still undergo α 1,3-fucosylation, which takes place in the medial-to-trans Golgi cisternae. This could tell whether the mutant proteins would fold correctly and export from the ER normally. (Figure 7- Figure 9)

8. Although from the confocal and mutagenesis analysis the authors proved the importance of LPYS motif in MNS3 for its proper Golgi localization, the underlying mechanism remains unresolved in this Ms. Why the LPYS is important for the Golgi retention? Any previously known mechanisms that could be related with COPII or COPI vesicles? Pull-down analysis using LPYS motif or MNS3 protein could be performed to dig out the interaction partners to address this important aspect for the Journal.

9. From the PNGase F assay (Figure 2C), MNS3 should be able to reach medial-to-trans Golgi cisternae, it may provide a hint that cis-Golgi retention MNS3 may due to the retrograde transport of the protein from trans-Golgi by COPI?

10. Most of the data presented in this Ms are derived from transient expression using tobacco leaf epidermal cell, some of the key data (such as the mutant LPYS affecting the Golgi retention) should be confirmed in transgenic plants.

11. Most of the data in the Ms using the chimeric construct (fuse with HDEL) or truncated construct (MNS3-CTS-GFP) to study the LPYS motif function in MNS3 Golgi retention. Why not use the full-length

MNS3? How about the full-length MNS3, will LPYS mutation affect Golgi localization of MNS3 in its native form?

Minor comments:

1. Different BFA treatment times (1h, 90min, 150min) are used in this study, why?
2. Figure 7 to Figure 9 are making the same conclusion that CTS is essential for MNS3 Golgi retention. It would be better if combine the figures and put the control to supplementary.
3. Quantification are really missing in this study. For example, the confocal results should be quantified for the percentage of Golgi and ER localization.
4. Figure 5E, a free GFP negative control might be necessary for the co-IP assay to ensure no non-specific interactions.
5. Figure 5F, it would be better to include loading control.
6. Figure 6C, it would be more consistent with the Figure 6B confocal image to show the western of MNS3-MNS1-GFP-HDEL to compare with the MNS1-MNS3-GFP-HDEL.
7. For the point mutation/deletion assay to pinpoint the essential motif for MNS3 localisation, the author may need to explain why the HDEL-tag form but not the native protein is used.

Reviewer #3 (Remarks to the Author):

This is an interesting paper in which the authors show that mannosidase 3 (MNS3), one of the enzymes involved in mannose trimming of the complex oligosaccharides on glycoproteins, is located in the cis-medial Golgi. Most of the other mannose trimming enzymes are located in the ER, so why not this one? The authors point out that if this enzyme was located in the ER then it would catalyze an oligosaccharide modification that would mistakenly target glycoprotein to the ERAD system. They then go on to show that MNS3 is not located in the ER and has a LPYS Golgi retention signal on its cytoplasmic tail.

Others have identified different cytoplasmic-tail Golgi retention signals, such as the KXD/E motif. However, the Golgi retention properties of MNS3 are unusual in that MNS3 is not resorbed into the ER when treating cells with Brefeldin. Even when the authors equip MNS3 with the HDEL signal on its luminal tail it fails to be resorbed by BFA.

The only shortcoming in the paper is that we don't know with what Golgi cytoplasmic component the LPYS signal interacts and the punctate structure to which MNS3 hangs on to when the rest of the cis-medial Golgi is resorbed following BFA treatment.

The paper is very well done and the figures showing the subcellular localization MNS3-GFP are very crisp and clear. There are only a few blips in the paper that need addressing:

Line 164 "that lack core α 1,3 fucosyltransferase activity

Line 180: The authors make the statement "All in all, the obtained data clearly point to a residency of MNS3 in the Golgi apparatus at steady-state, where it exists in an N-glycosylated form that is typically generated in the medial-to-trans-half of Golgi stacks." But isn't the situation more complex than that? Would it not be better to say "All in all, the obtained data indicate that MNS3 is predominantly located in the cis-medial Golgi apparatus at steady-state, although in an N-glycosylated form that is typical of form resident in the medial-to-trans-half of Golgi stacks.

Line 199. Authors cannot use the statement "data not shown." They need to show the data in a supplemental figure

Line 205. Why wasn't MNS3-GFP located in the ER after having blocked ER-to Golgi transport by a GTP-locked version of SAR1? If the authors used a GTP-locked version of SSAR1, then why doesn't it say so in the Figure legends?

Line 282 The authors conclude that appending HDEL sequence onto MNS3 and finding that it still goes to punctate structures indicates that the mechanism of HDEL-mediated ER retrieval is overruled by another, yet unknown mechanism. However, isn't possible that the structure of MNS3 simply makes the

HDEL sequence inaccessible to the HDEL receptor?

Fig 2A Order the panels Col-0, *mns3*, *mns3* MNS3::MNS3-GFP

Fig 2B does not show more than 1C. Put in supplemental figures and show with a Golgi marker.

Fig 5B The labeling on this figure does not make sense with respect to the information in the legend.

Fig. 9E The model needs more explanation in the figure legend

Reviewer #4 (Remarks to the Author):

This manuscript thoroughly re-examines the localization of Arabidopsis MNS3 (previously reported by the same authors as located in Golgi bodies of plant cells), which - supported by domain swap constructs with MNS1 - pointed to the CTS region. By N-terminal truncations and point mutations, the authors then identified a novel retention signal within the cytosolic region of the protein that confers steady state localization at the cis-most face of mobile Golgi stacks. Importantly, the Leucin (L) within the identified LPYs motif seems necessary and sufficient to retain MNS3 at BFA-resistant structures, similar to the Golgin CASP and SNARE Syp31 (and a handful of other transmembrane proteins showing the same behaviour). This strongly supports that plant cells also contain an ER-Golgi inter-mediate compartment (ERGIC), through which residents of both the ER and Golgi stacks cycle.

While the manuscript is well written and structured, the findings lack full potential of interpretation. For example, a complete overlap of MNS3 with Syp31 (Figure S1), but not with MNS1 or GnTI fusions, strongly supports spatial separation from both the ER and cis/medial Golgi cisternae. Unfortunately, there is also no explanation offered for the different Endo H-resistant patterns of MNS3-XFP-HDEL (single high MW band) versus MNS3-XFP (showing an additional discrete lower MW band) on immuno-blots (Figures 5F, 6C). If this is no mislabelling, this may indicate that ERD2-retrieved MNS3-XFP-HDEL versions do not - but MNS-XFP versions do - cycle through/come in contact with trimming and modification enzymes that are residents of Golgi bodies. And this is also reflected by the two fully modified GN-terminated N-glycan antenna on MNS3, probably due to continuous COPI-based retrograde vesicle traffic at continuous cisterna maturation from cis to trans (Day et al. 2013). Moreover, a possible reason for persistence of the punctate pattern upon BFA treatment is not speculated on (no comment on cytoskeleton depolymerizing drugs, although according to the M&M

section, actin-depolymerizing Latrunculin B was occasionally used to slow down Golgi movement). Finally, another role for the special localization of MNS3 in sites neither belonging entirely to the ER nor Golgi bodies should be given on evolutionary terms, since misdirection of MNS3 to the ER (fused to the CTS of GCSI) enhanced ERAD of mutant protein Bri1-5, which the authors previously showed to be governed by MNS4,5 (Hüttner et al. 2014). Thus in Arabidopsis wildtype (and other plant species sharing the LPY motif in MNS3), folding-compromised proteins like Bri1-5 are allowed to ESCAPE a (principally possible) more stringent degradation control in the ER, and partially reach the plasma membrane – which should be discussed based on the situation in yeast and mammalian cells.

Specifically:

The mutant allele of BRI1 (*bri1-5*) is retained in the ER due to misfolding (with unaffected N-glycan acquisition - but prevented cysteine-bridge formation within the N-terminal cap domain; Nogouchi et al. 1999). This, together with the underlying mutation in *bri1-5* (Cys69Tyr exchange), should be mentioned somewhere in the introduction, perhaps when citing Hong et al. Plant Cell (2007) for way of ERAD in *bri1-5* mutant plants. Especially since the lectin chaperons (CNX, CRT) had no major effect, but mutation of UGGT or BiP, hence identified as major factors for ER retention of the Bri1-5 proteins. Also, since being a proteasome-independent ERAD substrate, Bri1-5 decay is likely only initiated by mannose trimming due to inability to fold correctly. Thus, MNS4,5 mark the luminal ectodomain for destruction, wherever this occurs in case of integral membrane proteins (likely the central vacuole).

The authors should carefully check the text, when referring to the “Golgi apparatus” (GA), comprising the entire compartment (i.e. cis-Golgi to TGN) or Golgi complex (equivalent to the GA?), as opposed to Golgi bodies/stacks (without discrete TGN), or the cis-most Golgi cisternae (equivalent to ERGIC, cisterna initiators, and GECCO - when referring to Golgi remnants upon BFA treatment).

Going through the manuscript, I would like to point out the following,

Title Page 1. Considering my suggestions (below) I would change it, at least to: “cis-Golgi”

Summary:

Page 3, line 51: (...) IN the Golgi complex (...);

Same page, line 54: rather “special” than “strong” Golgi retention signal.

Background:

Same page, line 79: Plural (...) mannose residueS from the N-glycan.

Page 4, line 99: “also” instead of “largely” found on ER-resident glycoproteins.

Page 5, line 127: (...), dictates its retention AT the Golgi, (...)

Results:

Same page, line 144: is this really a good overlap? To my eyes, there is only partial overlap with MNS1 (albeit more than with ST) but much better i.e. perfect overlap with Syp31 (Figure S1).

Page 6, line 179: you may want to emphasize (here or later in the Discussion) that full processing to GN-terminated antennae indicates contact to all modification enzymes occurring in Golgi stacks.

Page 7, line 210ff: why do the authors cite ERES references when talking about a Golgi-confined protein (CASP) of plant cells? Structural studies by electron microscopy revealed earlier that the cis-most cisternae often stain much less intensely than medial and trans-Golgi cisternae, and this staining pattern correlated with the restricted localization of glycosylation and biosynthetic enzymes as compared to medial and trans-Golgi cisternae (see Donohoe et al. 2013 Traffic; Day et al. 2013 Histochem. Cell Biol.).

Page 8, line 220: any comment on possible involvement of the cytoskeleton/or linked component?

Same page, lines 223-237: the perfect overlap of MNS3 with Syp31 (Figure S1) indicates that Golgi remnants upon BFA treatment are equivalent to GECCO. I would move that Figure into the main.

Same page, lines 246-247, ditto. (...) remain on membrane STRUCTURES that are left behind (...) and are termed Golgi remnants/GECCO.

Page, line 256: Please state whether entire MNS1, MNS2, MNS3 reporter fusions were used here as opposed to a CTS fusion in case of ST-YFP (then it would be better to refer to the marker as STmd, as in Osterrieder et al. 2017).

Same page, line 268: (...) MAY form heteromers.

Same page, lower paragraph: please explain the different banding pattern of MNS3-XFP +/- HDEL tag!

Why does the HDEL version migrate as single large band? (due to ER-type high Man N-glycans and no contact with MNS1,2?), as compared to the MNS3-reporter version (showing a prominent second band, perhaps due to trimmed N-glycans (Man5GN2?). Please indicate MW standard bands in kDa!

Same page, line 282: I disagree – this statement is only correct considering localization. The different pattern on immunoblots indicates that the HDEL-tagged versions are bound by ERD2 also traveling through this site (ERGIC), but not to MNS1,2 harbouring cisternae (confirming spatial separation).

Page 12, line 356: why should that be „uncontrolled“? As shown before (Hüttner et al. 2014), MNS4 and MNS5 dominate the ERAD, despite ER residents with Man8GN2 structure generated by MNS3.

Same page, lines 364-365: This is not only the case in bri1-5, ...

Same page, last para: apparently the entire plant ERAD system is leaky, due to the special location of MNS3, allowing escape of partially malformed proteins!

Same page, lines 376-382: I do not agree to the conclusions, unless the authors conduct a back-cross to wildtype and analyse the wild-typical offspring expressing these transgenes for comparison.

Discussion

Page 13, line 397: the “assembly line model” assures efficiency, but the results are not in agreement with the full processing of MNS3 by all Golgi stack enzymes (remaining in the Δ XF background). In my view this rather supports the "cisterna maturation-based three stage model of the Golgi“ with continuous retrograde transport by COPII vesicles from trans to cis (Day et al. 2013 Traffic).

Same page, para on COPI: I quickly checked (in ARAMEMNON and ARAPORT), and found similar motifs (LPYi/r) in the 4 splice variants of a COPI/II associated p24 membrane protein (At5g01010) and interaction partner of AGB1 (At4g34460), a putative G-beta component of the canonical heterotrimeric G-protein complex. The authors may want to state that in their text.

Same page, lines 414-415: instead of “Golgi complex” I would rather refer to all possible terms in the literature here (i.e. ERGIC/cisterna initiators/GECCO/Golgi remnants).

Page 14, lines 423-424: reconsider Golgi residency, when the HDEL tag successfully prevented the low MW band of MNS-XFP fusions on immunoblots!

Same page: lines 428-430: ...or structural determinants, like the cytoskeleton or Golgi "glue" surrounding Golgi stacks in plant cells.

Same page, line 434-438: not all proteins, but those involved in N-glycan trimming/modification (Day et

al. 2013). This is not an entirely NEW proposition, considering the earlier structural studies (ERGIC, cisternal initiators) and Ito et al. (2018), who recently coined “GECCO”.

Same page, line 442: After all, this interface is not so “ill defined”, considering the above mentioned structural studies of 2013.

Same page, lines 444-450 meaning all proteins that travel through this cis-most compartment (ERGIC in other systems). Why would that be the case for ER residents – if reflecting passing through that interface, if not protecting them from further trimming by MNS4,5 in the ER?

Same page, lines 451-452: Really - interference with ERAD? You show that ER mislocalization of a GCS1-MNS3 fusion enhances ERAD of Bri1-5 by MNS4,5 – if this is not due to the CTS region of GCS1... (MNS3 plays no major role; Hüttner et al. 2014).

Page 15, line 471: cycles through (not to) the MNS3 compartment (...)

Same page, last para: Rather LPY is a newly identified “special signal” for accumulation in an intermediate layer (cis-most Golgi or ERGIC, neither entirely ER nor Golgi body yet), possibly due to interaction with a scaffold component (comment on cytoskeleton depolymerization drugs?)

Material & Methods

Page 18-19, lines 587-589: What means “If necessary, leaf segments were treated with the actin-depolymerizing agent latrunculin B (Calbiochem, stock solution at 1 mM in dimethyl sulphoxide) at a concentration of 25 μ M for 45 to 60 min prior to image acquisition to inhibit Golgi movement.” This is an important information and should be mentioned in the context of BFA-resistant Golgi remnants.

Figures

Please indicate MW sizes at the side of all immunoblots for better orientation.

References

Please add those of the structural studies (Day et al. and Donohue et al. from 2013).

Reviewers' comments:

Reviewer #1 (Remarks to the Author):

In the submitted paper, Schoberer and collaborators studied the Golgi retention mechanism of the Arabidopsis ER alpha-mannosidase I (MNS3). Using different constructs containing the MNS3 protein fused with fluorescent reporter and drugs provoking the relocation of Golgi enzymes, they reexamine the subcellular localization of the full-length MNS3 and demonstrate that MNS3 resides exclusively in the Cis/medial-Golgi cisternae and that the specific Golgi retention is due to the presence of a tetrapeptide signal in the cytoplasmic tail of the Arabidopsis MNS3. While the work is well done and the paper written in a comprehensive manner, I feel that the results remain interesting for a subset of scientists which are in the plant glycobiology community.

We thank the reviewer for the generally positive evaluation of our work. Our work provides novel insights into the organization of the early secretory pathway and should therefore be of relevance for the whole cell biology community. Moreover, the ER-associated degradation (ERAD) pathway is part of a highly conserved protein quality control process in yeast, mammals and plants and its correct functioning is critical for the wellbeing of organisms. In addition, due to the involvement of the mammalian MNS3 homologue in ERAD and the discrepancy around its subcellular localization, we are convinced that the results of our work are of interest for a wider community than only the plant glycobiology field.

Did the authors try to express a shorter tagged version of the MNS3 (e.g. c-myc; HA) in parallel to the fusion to the GFP? It would be worth proving that the behavior that they observed for the MNS3-GFP is not a special one due to the length of the MNS3-GFP and remains with a shorter tagged version. Did you try to express the MNS3-GFP in a stable manner in BY2 cells for example and observed the same behavior? Did they express the MNS3-HA-HDEL to confirm their results? Indeed, previous studies have shown that fusion with GFP induced abnormal subcellular localization and behavior of proteins.

We perfectly understand the reviewer's concerns. We are aware that great care has to be taken when working with recombinant proteins tagged with, for example, green fluorescent protein. However, we can rule out the possibility that MNS3-positive punctate structures represent unspecific artifacts of expressed protein for the following reasons. First, the full-length MNS3 protein lacking a critical, single leucine residue (L5A, leucine substituted with alanine) fused to GFP does not label any punctate structures after BFA treatment, which is in clear contrast to the behavior of the wildtype MNS3-GFP protein, although only a single amino acid has been exchanged. Instead, the fusion protein MNS3-L5A shows the "common" response to BFA and is fully redistributed to the ER like most other Golgi membrane proteins. Second, MNS3-labelled punctate structures are mobile and remain closely associated with the ER network. A similar observation has been made in other studies, using AtSYP31 and AtCASP. Also, mRFP-tagged MNS3-fusion proteins behave similarly to the GFP-tagged fusion proteins.

We have carried out most of our experiments in two different expression systems (transient expression in *N. benthamiana* and stable expression in *A. thaliana*) and obtained similar results. We do not see an obvious reason or advantage of using another expression system.

When they analyzed their images and mentioned respectively 9 and 12 images, is these images came from 9 and 12 independent plants? It would be good for the reader to clarify how many independent plants have been studied for the calculation of the Pearson's correlation coefficient.

We have now included more explanation in the main text and the respective materials & methods section to clarify this ambiguity. Briefly, we obtained 9 and 12 confocal images from 2 independent plants (from 1 infiltration event) expressing MNS3-GFP/MNS1-mRFP and MNS3-GFP/STtmd-mRFP, respectively.

When the authors performed the experiments with the BFA treatment, did they try to time course and different BFA concentrations? Dis they observe the same behavior for the MNS3-GFP? How do they make sure that the MNS3 does not need more time to relocate in the ER?

We added Suppl. Figure 1 that shows the BFA treatment of transiently co-expressed MNS3-GFP and STtmd-mRFP (as a Golgi reference marker) in *N. benthamiana* leaves, which have been treated with 50 µg/ml, 100 µg/ml or 200 µg/ml BFA and were imaged on a confocal microscope for up to 4 hours. Also, MNS3-GFP was expressed at a higher optical density (the oD600 was 0.15 instead of 0.10) to increase protein expression levels and visualize a potential localization in the ER. The experiment revealed that already after 1 hour of treating with half of the commonly used BFA concentration (which is 100 µg/ml unless stated otherwise) STtmd-mRFP has been fully relocated to the ER, whereas MNS3 largely remained on punctate structures even after longer incubation times. Longer incubation times, especially with high BFA concentrations are in our opinion not suitable as BFA inhibits protein secretion, which ultimately leads to cell death.

Reviewer #2 (Remarks to the Author):

In this manuscript, Schoberer et al. first re-visited the cis-Golgi sub-cellular localisation of steady-state MNS3 via confocal microscopy through transiently over-expression in *N. benthamiana* leaves and native-promoter-driven expression in *A. thaliana* mutants. Then the authors further defined the cis/medial-Golgi localisation of MNS3, and demonstrated its N-glycosylation in medial/trans-Golgi. Schoberer et al. proved that unlike the majority of Golgi-localised proteins, MNS3 is insensitive to BFA treatment or to the GTP-locked form of SAR1 (SAR1-GTP), revealing its specific localisation in Golgi remnants possibly originating from incomplete disassembly of Golgi stacks upon the disruption of early secretory pathway. Golgin AtCASP and cis-Golgi protein SYP31 were proved to display similar pattern. In addition, the authors used HDEL-tagged MNS3 (MNS3-GFP-HDEL) and its mutation forms and characterised that the fifth leucine residue (L5) in the MNS3 cytoplasmic tail is essential for MNS3 Golgi localisation. The role of MNS3's right location for ERAD degradation of BRI is also studied here in vivo.

In summary, this manuscript provides novel evidences of the crucial amino acid L5 form LPYS motif in MNS3 cytoplasmic tail for determining its specific cis-Golgi localisation, and the importance of this localisation in ERAD degradation of BRI. However, this manuscript offered little information on the underlying mechanisms of how the L5 possibly affect the MNS3 localisation. Two hypothesis exist for the important function of the leucine residue or the LPYS motif in MNS3: 1. Does it affect MNS3 ER

export? and 2. Does it facilitate MNS3 ER retrieval from cis-Golgi? If the answer is yes, further experiments could address the underlying mechanisms and thus the novelty of this study.

We thank the reviewer for the positive comments. We can comment that our data show that the LPYS motif is involved in Golgi retention and not in ER export or ER-retrieval (MNS3 lacking the motif is still present in the Golgi), but we do not know the interacting protein(s) or determinant(s) that retain it there. We have performed a preliminary proteomics approach to identify MNS3 interacting proteins. The approach resulted in a list of potential candidates that need to be validated for a role in MNS3 localization. Consequently, we are hesitant to conclude anything related to the underlying mechanism.

To investigate the possibility that COPI subunits are involved in the Golgi localization of MNS3, we transiently co-expressed MNS3-GFP as well as MNS3-102-GFP lacking the LPYS motif with RNAi constructs that silence endogenous δ COP and ϵ COP in *N. benthamiana* leaves. Interestingly, the Golgi localization of both the wildtype and the truncated MNS3 fusion protein was partially shifted to the vacuole in the presence of δ COP or ϵ COP RNAi constructs (Suppl. Figure 9). While this finding suggests an involvement of COPI-mediated recycling from the *trans*- to *cis*-Golgi, it does not explain the observed differences between the MNS3 variants.

Major comments:

1. How could the authors tell the puncta shown in Figure 2B is Golgi apparatus without co-localizing with the known Golgi markers? To assure the Golgi localization of the MNS3-GFP under native promoter, crossing or labeling with known Golgi markers are needed. (Figure 2B)

The reviewer is correct to point out this ambiguity. Hence, we crossed the Arabidopsis line *mns3* MNS3::MNS3-GFP with the recently published knockout mutant *gnt1* UBQ10::AtGnTI-mRFP (Schoberer et al., Plant Physiol 2019; doi: 10.1104/pp.19.00310). Arabidopsis GnTI is a *bona fide* Golgi marker that has been used in numerous studies and co-localizes very well with MNS3 (see Figure 2B).

2. It is wired that when co-expressing of SAR1-GTP with MNS3-GFP and ST-mRFP, only ST-mRFP will be trapped in the ER but MNS3 remains puncta localization. Since MNS3 is synthesized in the ER and delivered from the ER to the Golgi apparatus via the COPII vesicles, why blocking the COPII-mediated ER export will not result in any ER pattern of MNS3? (Figure 3B)

It is indeed an unusual behavior for glycosylation-related enzymes, but other proteins like CASP and SYP31 behave in a similar manner. We would like to point out that expression levels of MNS3 are relatively low in comparison to other previously studied Golgi-resident glycosylation enzymes. At higher MNS3 expression levels, the presence of MNS3 in the ER in response to blocking ER export with the dominant-negative mutant SAR1-GTP can indeed be observed. Hence, we have replaced the respective SAR1-GTP images in Figure 3B that show some accumulation of MNS3-GFP in the ER when expressed at higher levels. Suppl. Figure 1 also shows some accumulation of MNS3-GFP in the ER in response to BFA.

3. When treated with BFA for 150 min, the authors claimed that MNS3 exhibited weak ER pattern

(Figure 3C). Nevertheless, it is hard to tell whether this is ER pattern or just some small puncta. ER markers should be used to confirm it. (Figure 3C)

We agree with the reviewer that the ER pattern very often is not clearly observable. In fact, the ER labelling is only observable at higher MNS3 expression levels that are generally difficult to achieve with MNS3. To fulfill the reviewer's request, we have expressed MNS3-GFP (with a higher optical density) together with the ER-resident protein DAD1 (Defender against Apoptotic Death-1; Danon et al., JBC 2004, doi: 10.1074/jbc.M304468200) fused to mRFP followed by a BFA treatment over 3 hours (see Suppl. Figure 8)

4. For the Co-IP assay, Free-GFP should be used as a negative control. (Figure 5E)

We thank the reviewer for this suggestion. We opted for a negative control specifically designed to validate the specificity of the GFP- and mRFP-Trap purification system, respectively (Suppl. Figure 14). We co-expressed the Golgi-resident MNS1-mRFP fusion protein with ER-localized GCSI-AtGnTI-GFP and co-purified the proteins using mRFP-Trap beads. As expected, MNS1-mRFP bound to mRFP-Trap beads, whereas GCSI-AtGnTI-GFP was not co-purified as no Western blot signal was detected. Similarly, MNS3-GFP was expressed alone or together with mRFP-HDEL and purified with GFP-Trap beads. Only MNS3-GFP bound to GFP-Trap beads, whereas no binding of mRFP-HDEL to GFP-Trap beads or MNS3-GFP was detected by immunoblotting.

5. For the Endo H digestion, no significant shift of MNS1-mRFP-HDEL was observed! Lower percentage of SDS-PAGE should be used to differentiate the shift. (Figure 5F)

A prediction of N-glycosylation sites in the AtMNS1 protein sequence by NetNGlyc 1.0 (<http://www.cbs.dtu.dk/services/NetNGlyc/>) reveals 4 putative N-glycosylation sites, two of them with a high probability. This means that an EndoH digestion will remove only a few kilo-Daltons, resulting in a tiny shift in protein mobility. We have repeated the experiment and replaced the previous figure, hopefully making the mobility shift of the MNS1-mRFP-HDEL protein band after EndoH treatment a little clearer.

6. Why the MNS3-mRFP showed two bands in Figure 5F? (Figure 5F)

The additional band is unspecific and generated during the extraction process. Fortunately, we were able to resolve this issue. We have changed the protein extraction protocol and repeated the EndoH digest and Western blot, which led to the disappearance of the faster migrating band of MNS3-mRFP. In detail, ground leaf material was not mixed with extraction buffer, but instead was directly mixed with 1.5x Laemmli buffer and boiled at 95°C for 5 minutes. Figure 5F has been replaced.

7. Deletion and mutation analysis showed that the N-terminal cytoplasmic tail of MNS3 contains a signal motif LPYS affecting the protein Golgi localization. Nevertheless, will the LPYS motif mutation affect normal ER export or proper protein folding of MNS3, resulting in the ER pattern? The authors could use the same assay in Figure 2 (PNGase F assay) to see whether the mutants still undergo α 1,3-fucosylation, which takes place in the medial-to-trans Golgi cisternae. This could tell whether the mutant proteins would fold correctly and export from the ER normally. (Figure 7- Figure 9)

We thank the reviewer for this great suggestion. We therefore expressed full-length versions of wildtype MNS3-GFP, mutated MNS3-L5A-GFP and truncated MNS3-102-GFP (lacking the N-terminal amino acids 2 to 8 with the LPYS motif of the cytoplasmic tail) under the control of the ubiquitin 10 promoter in *N. benthamiana* wildtype and Δ XF plants (which have reduced or no α 1,3-fucosylation) and subjected total protein extracts to PNGase F and Endo H treatments. SDS-PAGE separation and immunoblotting with antibodies against GFP revealed an electrophoretic mobility shift of all three PNGase F-treated MNS3 fusion proteins isolated from Δ XF plants with strongly reduced core α 1,3-fucosyltransferase activity, whereas EndoH-treated samples did not show any shifts. In addition, no mobility shift was detected of either PNGase F- or EndoH-treated MNS3 fusion proteins isolated from wildtype plants that still produce complex N-glycans with core α 1,3-fucose residues. This result indicates that both the wildtype MNS3 as well as mutated MNS3-L5A and truncated MNS3-102 fusion proteins are competent for ER export and capable of successfully travelling to the medial/*trans*-Golgi, where they undergo α 1,3-fucosylation (see Suppl. Figure 11).

8. Although from the confocal and mutagenesis analysis the authors proved the importance of LPYS motif in MNS3 for its proper Golgi localization, the underlying mechanism remains unresolved in this Ms. Why the LPYS is important for the Golgi retention? Any previously known mechanisms that could be related with COPII or COPI vesicles? Pull-down analysis using LPYS motif or MNS3 protein could be performed to dig out the interaction partners to address this important aspect for the Journal.

We absolutely agree with the reviewer's idea. We have tried to identify so far unknown interaction partners of MNS3 from its native cellular environment by subjecting whole-seedling protein extracts from the Arabidopsis *mns3* MNS3::MNS3-GFP line to a one-step GFP-based co-purification using GFP-Trap beads (anti-GFP antibodies coupled to agarose beads, Chromotek) that will pull down MNS3-GFP along with bound proteins (Schoberer et al., Plant Physiol 2013, doi: 10.1104/pp.112.210757; Hüttner et al., Biochem J 2014, doi: 10.1042/BJ20141057). Eluted proteins were pelleted and separated by SDS-PAGE. Coomassie-stained protein bands were excised from the gel and trypsin-digested. The peptides were analyzed by LC-ESI-MS/MS and identified using a MS/MS search engine and the SwissProt database. Our proteomics approach has yielded a couple of ER and Golgi proteins, some of them with known (regulatory) functions in cargo selection and transport. These proteins were only identified in the *mns3* MNS3::MNS3-GFP line and hence could be potential MNS3 interaction partners. The list also included a number of uncharacterised proteins that according to *in silico* analysis potentially enter the secretory pathway and hence are also interesting candidates. In future experiments, we want to use synthetic peptides of the N-terminal amino acids of the cytoplasmic MNS3 tail, either with or without the LPYS motif, to fish for COPI or COPII subunits.

9. From the PNGase F assay (Figure 2C), MNS3 should be able to reach medial-to-trans Golgi cisternae, it may provide a hint that cis-Golgi retention MNS3 may due to the retrograde transport of the protein from trans-Golgi by COPI?

We agree with the reviewer. To investigate the possibility that COPI subunits are involved in the Golgi localization of MNS3, we transiently co-expressed MNS3-GFP as well as UBQ10::MNS3-102-GFP (lacking the LPYS motif) with RNAi constructs that silence endogenous δ COP and ϵ COP in *N. benthamiana* leaves (as described recently in Schoberer et al., Plant Physiol 2019; doi: 10.1104/pp.19.00310). Interestingly, the Golgi localization of both the wildtype and the truncated MNS3 fusion protein was partially shifted to the vacuole in the presence of δ COP or ϵ COP RNAi

constructs. This finding suggests that both Golgi-resident MNS3 fusion proteins are not efficiently recycled from the *trans*-Golgi cisternae when COPI formation is impaired, which leads to vacuolar targeting. This observation is in agreement with earlier studies investigating the Golgi protein EMP12 (Gao et al., Trends Plant Sci 2014, doi: 10.1016/j.tplants.2014.04.004; Woo et al., Mol Biol Cell 2015, doi: 10.1091/mbc.E15-06-0361). Notably, no obvious differences in the response of the wildtype versus the truncated MNS3 fusion protein towards δ COP or ϵ COP silencing were observed, which suggests that the LPYS motif may not be required for COPI binding. This experiment has been added as Supplemental Figure 9.

10. Most of the data presented in this Ms are derived from transient expression using tobacco leaf epidermal cell, some of the key data (such as the mutant LPYS affecting the Golgi retention) should be confirmed in transgenic plants.

To remedy this shortcoming, we generated the Arabidopsis line *bri1-5* UBQ10::MNS3-L5A-GFP and treated leaves of 8-days-old seedlings with 100 μ g/ml BFA. As a comparison, we performed a similar experiment with *bri1-5* UBQ10::MNS3-GFP seedlings in parallel (Suppl. Figure 13). UBQ10::MNS3-GFP remained on punctate membrane structures after 2h BFA treatment, whereas UBQ10::MNS3-L5A-GFP was visibly relocalized to the ER. This figure confirms the results from our experiments obtained from the transient expression of the same fusion proteins in *N. benthamiana* leaves (Figure 8).

11. Most of the data in the Ms using the chimeric construct (fuse with HDEL) or truncated construct (MNS3-CTS-GFP) to study the LPYS motif function in MNS3 Golgi retention. Why not use the full-length MNS3? How about the full-length MNS3, will LPYS mutation affect Golgi localization of MNS3 in its native form?

We have generated a new construct for the expression of MNS3-102-GFP, which is a full-length version of MNS3 (CTS region plus catalytic domain) lacking the N-terminal amino acids 2 to 8 of the cytoplasmic tail (lacking the LPYS motif) under the control of the ubiquitin 10 promoter. We have transiently expressed this fusion protein in parallel with UBQ10::MNS3-GFP and UBQ10::MNS3-L5A-GFP in *N. benthamiana* leaves, treated leaves with BFA and observed effects on localization by confocal microscopy. The full-length fusions behaved similar to BFA as the truncated CTS fusion proteins. Briefly, only the wildtype UBQ10::MNS3-GFP fusion protein continued to label punctate structures upon BFA treatment, whereas the mutated UBQ10::MNS3-L5A-GFP and truncated UBQ10::MNS3-102-GFP fusion proteins were completely shifted to the ER. A figure summarizing this result now replaces the previous figure (Figure 8), which showed the localization of the truncated MNS3-CTS-GFP fusion proteins before and after BFA treatment and in this version of the manuscript is shown as Suppl. Figure 5.

Minor comments:

1. Different BFA treatment times (1h, 90min, 150min) are used in this study, why?

Throughout our work, we have performed time-course experiments during which we have imaged cells at least every 60 minutes. For the ease of presentation and reading, we have presented images

that are most representative for the experiment and as clear as possible, even though they are from different time points. Nonetheless, we have replaced the respective BFA images in Figure 3 with images showing the localization of the respective fusion proteins after 2 hours BFA treatment.

2. Figure 7 to Figure 9 are making the same conclusion that CTS is essential for MNS3 Golgi retention. It would be better if combine the figures and put the control to supplementary.

Figure 8 showing the localization of the mutated MNS3-CTS fusion proteins has been moved to the Supplemental section (Suppl. Figure 5) and was replaced with a figure that shows the localization of wildtype, mutated and truncated full-length MNS3-GFP before and after BFA treatment.

3. Quantification are really missing in this study. For example, the confocal results should be quantified for the percentage of Golgi and ER localization.

From our point of view, the quantification of ER localization in plant cells is simply impossible. Leaf epidermal cells, for example, differ greatly in shape and size (or volume). The number of Golgi stacks per cell is not uniform, either. It would be futile to compare Golgi versus ER localization between different cells as numbers would not be meaningful for the above mentioned reasons.

4. Figure 5E, a free GFP negative control might be necessary for the co-IP assay to ensure no non-specific interactions.

See Major Comments, point 4.

5. Figure 5F, it would be better to include loading control.

We did not include a loading control, because we do not compare protein concentrations and instead aim to show potential mobility shifts due to removal of glycans.

6. Figure 6C, it would be more consistent with the Figure 6B confocal image to show the western of MNS3-MNS1-GFP-HDEL to compare with the MNS1-MNS3-GFP-HDEL.

The reviewer is right. We have generated the respective data and replaced the previous figure with a new figure comparing EndoH-digested MNS3-MNS1-GFP-HDEL with MNS1-MNS3-GFP-HDEL.

7. For the point mutation/deletion assay to pinpoint the essential motif for MNS3 localisation, the author may need to explain why the HDEL-tag form but not the native protein is used.

By tagging with the HDEL retrieval sequence, an immediate effect of the introduced deletions/point mutations on MNS3 localization can be observed. A localization of a MNS3 variant in the ER is the immediate result of HDEL-mediated retrieval of the fusion protein from the Golgi to the ER, indicating that the introduced deletion or mutation abolishes retention of MNS3 in the Golgi apparatus.

Reviewer #3 (Remarks to the Author):

This is an interesting paper in which the authors show that mannosidase 3 (MNS3), one of the enzymes involved in mannose trimming of the complex oligosaccharides on glycoproteins, is located in the cis-medial Golgi. Most of the other mannose trimming enzymes are located in the ER, so why not this one? The authors point out that if this enzyme was located in the ER then it would catalyze an oligosaccharide modification that would mistakenly target glycoprotein to the ERAD system. They then go on to show that MNS3 is not located in the ER and has a LPYS Golgi retention signal on its cytoplasmic tail.

Others have identified different cytoplasmic-tail Golgi retention signals, such as the KXD/E motif. However, the Golgi retention properties of MNS3 are unusual in that MNS3 is not resorbed into the ER when treating cells with Brefeldin. Even when the authors equip MNS3 with the HDEL signal on its luminal tail it fails to be resorbed by BFA.

The only shortcoming in the paper is that we don't know with what Golgi cytoplasmic component the LPYS signal interacts and the punctate structure to which MNS3 hangs on to when the rest of the cis-medial Golgi is resorbed following BFA treatment.

The paper is very well done and the figures showing the subcellular localization MNS3=GFP are very crisp and clear. There are only a few blips in the paper that need addressing:

We thank the reviewer for these positive comments.

Line 164 "that lack core α 1,3 fucosyltransferase activity

This sentence has been changed according to the reviewer's suggestion.

Line 180: The authors make the statement "All in all, the obtained data clearly point to a residency of MNS3 in the Golgi apparatus at steady-state, where it exists in an N-glycosylated form that is typically generated in the medial-to-trans-half of Golgi stacks." But isn't the situation more complex than that? Would it not be better to say "All in all, the obtained data indicate that MNS3 is predominantly located in the cis-medial Golgi apparatus at steady-state, although in an N-glycosylated form that is typical of form resident in the medial-to-trans-half of Golgi stacks.

The presence of oligo-mannosidic structures (like Man8) attached to the glycoprotein is a hallmark of retention in the ER, while forward movement to the Golgi will lead to processing of N-glycans and the formation of complex N-glycans carrying β 1,2-xylose and core α 1,3- fucose residues (main glycoform: GnGnXF). *N. tabacum* N-acetylglucosaminyltransferase I (GnTI) resides in cis/medial-Golgi cisternae and, like MNS3, predominantly carries GnGnXF structures (Schoberer et al., Traffic 2009, doi: 10.1111/j.1600-0854.2008.00841.x.; Plant J 2014, doi: 10.1111/tpj.12671). By using fluorescence recovery after photobleaching (FRAP), it has been shown by Brandizzi et al. (Plant Cell 2002, doi: 10.1105/tpc.001586) and our lab (Schoberer et al., Traffic 2010, doi: 10.1111/j.1600-0854.2010.01106.x) that Golgi-resident proteins, such as STTmd-GFP, AtERD2-GFP, GnTI, MII or GALT1 are not stably associated with the Golgi complex and instead undergo continuous movement in and out of Golgi stacks thereby receiving processing also by medial/trans-Golgi located N-glycan processing enzymes resulting in the generation of complex N-glycans. At the trans-Golgi, these proteins are then thought to be recycled back to their steady-state location in the Golgi by largely unknown mechanisms, possibly involving a COPI-complex mediated retrieval mechanism. In the main text of the manuscript, we have now simplified our statement to avoid any confusion on the reader's side.

Line 199. Authors cannot use the statement "data not shown." They need to show the data in a supplemental figure.

We agree with the reviewer. We have added a figure that shows the BFA treatment of MNS3-GFP and STtmd-mRFP over 3 hours (Suppl. Figure 1B).

Line 205. Why wasn't MNS3-GFP located in the ER after having blocked ER-to Golgi transport by a GTP-locked version of SAR1? If the authors used a GTP-locked version of SAR1, then why doesn't it say so in the Figure legends?

Expression levels of MNS3 are relatively low in comparison to other previously studied Golgi-resident glycosylation enzymes. At higher MNS3 expression levels, however, the presence of MNS3 in the ER in response to blocking ER export with the dominant-negative mutant SAR1-GTP can be observed. Hence, the respective image panel in Figure 3B showing MNS3-GFP co-expressed with STtmd-mRFP and SAR1-GTP has been replaced with new images where some accumulation of MNS3 in the ER is observable. The figure legend does state that SAR1-GTP was used.

Line 282 The authors conclude that appending HDEL sequence onto MNS3 and finding that it still goes to punctate structures indicates that the mechanism of HDEL-mediated ER retrieval is overruled by another, yet unknown mechanism. However, isn't possible that the structure of MNS3 simply makes the HDEL sequence inaccessible to the HDEL receptor?

The behavior of the MNS1-MNS3-GFP-HDEL-tagged protein shows that MNS3-GFP-HDEL is accessible to the HDEL receptor. The fusion proteins containing short deletions or a single amino acid substitution in the cytoplasmic tail of MNS3 were specifically redistributed to the ER, similar to HDEL-tagged MNS1 or ST (Silva-Alvim et al., Plant Cell 2018; doi: 10.1105/tpc.18.00426) providing additional evidence for access to the HDEL receptor. However, we cannot completely exclude that binding to the LPYS motif on the cytosolic side of the membrane causes a conformational change of the luminal MNS3 protein portion that prevents the access to the HDEL receptor.

Fig 2A Order the panels Col-0, *mns3*, *mns3* MNS3::MNS3-GFP

The panels have been swapped around according to the reviewer's suggestion.

Fig 2B does not show more than 1C. Put in supplemental figures and show with a Golgi marker.

We crossed the Arabidopsis line *mns3* MNS3::MNS3-GFP with the recently published knockout mutant *gnt1* UBQ10::AtGnTI-mRFP (Schoberer et al., Plant Physiol 2019; doi: 10.1104/pp.19.00310). Arabidopsis GnTI is a *bona fide* Golgi marker (Grebe et al. 2003, Current Biology, DOI 10.1016/S0960-9822(03)00538-4) and co-localizes very well with MNS3 (see Figure 2B). We prefer to leave this figure in the main text, though.

Fig 5B The labeling on this figure does not make sense with respect to the information in the legend.- We agree with the reviewer. The title of the figure legend has been corrected.

Fig. 9E The model needs more explanation in the figure legend

We have now added a more detailed description of the model.

Reviewer #4 (Remarks to the Author):

This manuscript thoroughly re-examines the localization of Arabidopsis MNS3 (previously reported by the same authors as located in Golgi bodies of plant cells), which - supported by domain swap constructs with MNS1 - pointed to the CTS region. By N-terminal truncations and point mutations, the authors then identified a novel retention signal within the cytosolic region of the protein that confers steady state localization at the cis-most face of mobile Golgi stacks. Importantly, the Leucine (L) within the identified LPYs motif seems necessary and sufficient to retain MNS3 at BFA-resistant structures, similar to the Golgin CASP and SNARE Syp31 (and a handful of other transmembrane proteins showing the same behaviour). This strongly supports that plant cells also contain an ER-Golgi inter-mediate compartment (ERGIC), through which residents of both the ER and Golgi stacks cycle.

While the manuscript is well written and structured, the findings lack full potential of interpretation. For example, a complete overlap of MNS3 with Syp31 (Figure S1), but not with MNS1 or GnTI fusions, strongly supports spatial separation from both the ER and cis/medial Golgi cisternae. Unfortunately, there is also no explanation offered for the different Endo H-resistant patterns of MNS3-XFP-HDEL (single high MW band) versus MNS3-XFP (showing an additional discrete lower MW band) on immuno-blots (Figures 5F, 6C). If this is no mislabelling, this may indicate that ERD2-retrieved MNS3-XFP-HDEL versions do not - but MNS3-XFP versions do - cycle through/come in contact with trimming and modification enzymes that are residents of Golgi bodies. And this is also reflected by the two fully modified GN-terminated N-glycan antenna on MNS3, probably due to continuous COPI-based retrograde vesicle traffic at continuous cisterna maturation from cis to trans (Day et al. 2013).

We thank the reviewer for the helpful comments. The additional band is indeed unspecific and generated during the extraction process.

Moreover, a possible reason for persistence of the punctate pattern upon BFA treatment is not speculated on (no comment on cytoskeleton depolymerizing drugs, although according to the M&M section, actin-depolymerizing Latrunculin B was occasionally used to slow down Golgi movement). Finally, another role for the special localization of MNS3 in sites neither belonging entirely to the ER nor Golgi bodies should be given on evolutionary terms, since misdirection of MNS3 to the ER (fused to the CTS of GCSI) enhanced ERAD of mutant protein Bri1-5, which the authors previously showed to be governed by MNS4,5 (Hüttner et al. 2014). Thus in Arabidopsis wildtype (and other plant species sharing the LPY motif in MNS3), folding-compromised proteins like Bri1-5 are allowed to ESCAPE a (principally possible) more stringent degradation control in the ER, and partially reach the plasma membrane - which should be discussed based on the situation in yeast and mammalian cells.

Regarding the use of latB, we have clarified this ambiguity now in the materials & methods section. We have also added additional information for ERAD of BRI1-5 to make it clearer.

Specifically:

The mutant allele of BRI1 (*bri1-5*) is retained in the ER due to misfolding (with unaffected N-glycan acquisition - but prevented cysteine-bridge formation within the N-terminal cap domain; Nogouchi et al. 1999). This, together with the underlying mutation in *bri1-5* (Cys69Tyr exchange), should be mentioned somewhere in the introduction, perhaps when citing Hong et al. Plant Cell (2007) for way of ERAD in *bri1-5* mutant plants. Especially since the lectin chaperons (CNX, CRT) had no major effect, but mutation of UGGT or BiP, hence identified as major factors for ER retention of the Bri1-5 proteins. Also, since being a proteasome-independent ERAD substrate, Bri1-5 decay is likely only initiated by mannose trimming due to inability to fold correctly. Thus, MNS4,5 mark the luminal ectodomain for destruction, wherever this occurs in case of integral membrane proteins (likely the central vacuole).

We thank the reviewer for pointing this out. We have added more information about *bri1-5* in the respective results section (“Predominant Golgi localization is important to prevent MNS3-mediated ERAD”).

The authors should carefully check the text, when referring to the “Golgi apparatus” (GA), comprising the entire compartment (i.e. cis-Golgi to TGN) or Golgi complex (equivalent to the GA?), as opposed to Golgi bodies/stacks (without discrete TGN), or the cis-most Golgi cisternae (equivalent to ERGIC, cisterna initiators, and GECCO - when referring to Golgi remnants upon BFA treatment.

In literature, the term “Golgi complex” is equivalent to the term “Golgi apparatus”, which includes the Golgi stack (*cis*-medial-*trans* cisternae) plus the TGN, whereas the term “Golgi stacks” describes the *cis*-medial-*trans* cisternae only. In our opinion, *cis*-most Golgi cisternae represent the C1 cisterna, possibly the C2 cisterna, of the maturing Golgi stack that face the ER and potentially may have an ERGIC-like nature in plant cells. To distinguish between the localization of MNS3 in the Golgi apparatus at steady-state and BFA-induced punctate structures, the latter are now referred to as membrane structures possibly similar to the ERGIC, *cis* initiators, GECCO or Golgi remnants as suggested by the reviewer. Also, we now only use the terms “Golgi apparatus” or “Golgi stacks” when referring to the steady-state localization of MNS3.

Going through the manuscript, I would like to point out the following,

Title Page 1. Considering my suggestions (below) I would change it, at least to: “cis-Golgi”

We have changed the title according to the reviewer’s suggestion.

Summary:

Page 3, line 51: (...) IN the Golgi complex (...);

We have corrected the sentence.

Same page, line 54: rather “special” than “strong” Golgi retention signal.

We have made the suggested changes. The phrase “strong Golgi retention signal” was changed into “specific Golgi retention signal”.

Background:

Same page, line 79: Plural (...) mannose residueS from the N-glycan.

We have corrected the mistake.

Page 4, line 99: “also” instead of “largely” found on ER-resident glycoproteins.

We have changed this sentence into “ER-resident glycoprotein contain large amounts of $\text{Man}_8\text{GlcNAc}_2$ oligosaccharides.

Page 5, line 127: (...), dictates its retention AT the Golgi, (...)

We have corrected the sentence.

Results:

Same page, line 144: is this really a good overlap? To my eyes, there is only partial overlap with MNS1 (albeit more than with ST) but much better i.e. perfect overlap with Syp31 (Figure S1).

We have changed the statement “good overlap” into “showed the colocalization of both proteins”.

Page 6, line 179: you may want to emphasize (here or later in the Discussion) that full processing to GN-terminated antennae indicates contact to all modification enzymes occurring in Golgi stacks.

Mass spectrometry has shown that MNS3 carries the Golgi-processed complex N-glycan structure GnGnXF. We cannot confirm that MNS3 is modified by the late-Golgi enzymes β 1,3-galactosyltransferase and α 1,4-fucosyltransferase that generate the Lewis a epitope. So, technically, we cannot claim that MNS3 gets in contact with all Golgi-resident modification enzymes.

Page 7, line 210ff: why do the authors cite ERES references when talking about a Golgi-confined protein (CASP) of plant cells? Structural studies by electron microscopy revealed earlier that the cis-most cisternae often stain much less intensely than medial and trans-Golgi cisternae, and this staining pattern correlated with the restricted localization of glycosylation and biosynthetic enzymes as compared to medial and trans-Golgi cisternae (see Donohoe et al. 2013 Traffic; Day et al. 2013 Histochem. Cell Biol.).

We specifically cite papers that explain the function of AtCASP as a *cis*-Golgi tether/matrix protein during Golgi biogenesis and describe its response to SAR1-GTP co-expression from a previous study, showing its accumulation on ERES. We are not citing papers that deal with ERES *per se*. The steady-state localization of AtCASP in the Golgi is also not the subject. The structural studies mentioned above are cited at another point in the manuscript.

Page 8, line 220: any comment on possible involvement of the cytoskeleton/or linked component? -

We are not completely sure what the reviewer is referring to. We can only comment that we mostly used cells with an intact cytoskeleton (no use of immobilizing chemicals, unless FLIM measurements or co-localization analyzes were conducted). Also, although the actin cytoskeleton is important for movement of plant secretory organelles, its disruption does not impair short-distance membrane flow in the early secretory pathway. For example, ER/Golgi protein exchange can occur in the absence of actin, as demonstrated by separate approaches in cells treated with actin-disrupting agents (i.e. Brandizzi et al., Plant Cell 2002, doi: 10.1105/tpc.001586).

Same page, lines 223-237: the perfect overlap of MNS3 with Syp31 (Figure S1) indicates that Golgi remnants upon BFA treatment are equivalent to GECCO. I would move that Figure into the main.
We agree that the high degree of co-localization between MNS3 and SYP31 is remarkable. Due to space restrictions, we would prefer to leave this figure in the supplemental section.

Same page, lines 246-247, ditto. (...) remain on membrane STRUCTURES that are left behind (...) and are termed Golgi remnants/GECCO.

We have changed the sentence according to the reviewer's suggestion.

Page, line 256: Please state whether entire MNS1, MNS2, MNS3 reporter fusions were used here as opposed to a CTS fusion in case of ST-YFP (then it would be better to refer to the marker as STtmd, as in Osterrieder et al. 2017).

We thank the reviewer for this suggestion. We are now referring to ST as "STtmd". We used full-length protein fusions unless stated otherwise (i.e. CTS or tmd fusion).

Same page, line 268: (...) MAY form heteromers.

Our lab has shown several times (Schoberer et al., Plant Physiol 2013, doi: 10.1104/pp.112.210757; Schoberer et al., Plant J 2014, doi: 10.1111/tpj.12671; this manuscript) by *in planta* FRET-FLIM and *in vitro* co-IP that many Golgi-resident N-glycan processing enzymes, especially those from *cis*/medial-Golgi cisternae, dimerize (homo- and heterodimerization).

Same page, lower paragraph: please explain the different banding pattern of MNS3-XFP +/- HDEL tag! Why does the HDEL version migrate as single large band? (due to ER-type high Man N-glycans and no contact with MNS1,2?), as compared to the MNS3-reporter version (showing a prominent second band, perhaps due to trimmed N-glycans (Man5GN2?). Please indicate MW standard bands in kDa!
The lower molecular weight band of MNS3-mRFP is an artefact from the extraction procedure. A change in the protein extraction protocol led to the disappearance of the faster migrating band of MNS3-mRFP (Figure 5F has been replaced with a new immunoblot). In detail, ground leaf material was not mixed with extraction buffer, but instead was directly mixed with 1.5x Laemmli buffer and boiled at 95°C for 5 minutes. Molecular weight standards are now given for all immunoblots.

Same page, line 282: I disagree – this statement is only correct considering localization. The different pattern on immunoblots indicates that the HDEL-tagged versions are bound by ERD2 also traveling through this site (ERGIC), but not to MNS1,2 harbouring cisternae (confirming spatial separation).

We rephrased the statement and now refer to a similarity of MNS3-GFP-HDEL and MNS3 with respect to their localization.

Page 12, line 356: why should that be „uncontrolled“? As shown before (Hüttner et al. 2014), MNS4 and MNS5 dominate the ERAD, despite ER residents with Man8GN2 structure generated by MNS3.

We have changed the phrase “uncontrolled degradation” into “disordered degradation”.

Same page, lines 364-365: This is not only the case in *bri1-5*, ...

We thank the reviewer for pointing this out. ERAD is regulated in wild-type and in *bri1-5*. This sentence has been rephrased accordingly.

Same page, last para: apparently the entire plant ERAD system is leaky, due to the special location of MNS3, allowing escape of partially malformed proteins!

Possibly, the ERAD system is not exactly leaky. ERAD substrates likely also cycle to the MNS3 compartment, because their N-glycans normally lack the terminal mannose residues on the B- and C-branch (see discussion), indicating processing by Golgi-resident MNS3 and ER-resident MNS4/5. Examples for such ERAD substrates are *bri1-5* (Hüttner et al., Plant Cell 2014, doi:10.1105/tpc.114.123216) or a misfolded variant of the STRUBBELIG (SUB) extracellular domain (SUBEX-C57Y; Hüttner et al., Biochem J 2014, doi:10.1042/BJ20141057). After MNS3-mediated processing, ERAD substrates as well as ER-residents are cycled back to the ER. It is possible that MNS3 together with important proteins such as AtSYP31, AtCASP etc is placed in a membrane compartment that serves as checkpoint for the separation of ERAD substrates and ER residents from secretory cargo proteins that receive further processing in the Golgi.

Same page, lines 376-382: I do not agree to the conclusions, unless the authors conduct a back-cross to wildtype and analyse the wild-typical offspring expressing these transgenes for comparison. To fulfill the reviewer's request, we have added Suppl. Figure 10, which shows images of *bri1-5* plants expressing MNS3::MNS3-L5A-GFP backcrossed with Col-0 wildtype (F1 generation, heterozygous) in comparison to *bri1-5* MNS3::MNS3-L5A-GFP and *bri1-5* soil-grown plants. As expected, the enhanced *bri1-5* phenotype of plants expressing MNS3::MNS3-L5A-GFP is lost in the backcrossed line. The immunoblot shows the expression of the fusion proteins in the respective lines. Tubulin detection was used as a control.

Discussion

Page 13, line 397: the "assembly line model" assures efficiency, but the results are not in agreement with the full processing of MNS3 by all Golgi stack enzymes (remaining in the Δ XF background). In my view this rather supports the "cisterna maturation-based three stage model of the Golgi" with continuous retrograde transport by COPII vesicles from trans to cis (Day et al. 2013 Traffic).

The assembly line model refers to the step-wise biosynthetic activities of N-glycan processing enzymes within the pathway. We are aware that Golgi-localized processing enzymes are no true residents of the Golgi and continually move in and out of the stack, which indeed requires their retrieval from *trans* to *cis* cisternae. The movement of enzymes through the entire stack has been confirmed in our lab via fluorescence recovery after photobleaching and N-glycan profiling experiments. Our data are consistent with the carbohydrate synthesis stage as shown by Day et al. and COPI-mediated recycling of glycan processing enzymes.

Same page, para on COPI: I quickly checked (in ARAMEMNON and ARAPORT), and found similar motifs (LPYi/r) in the 4 splice variants of a COPI/II associated p24 membrane protein (At5g01010) and interaction partner of AGB1 (At4g34460), a putative G-beta component of the canonical heterotrimeric G-protein complex. The authors may want to state that in their text

The two mentioned proteins contain a partially conserved LPYS motif. Several of such proteins can be found in the *A. thaliana* protein database, but an involvement in a similar process is not obvious. A

Patmatch search using the “LPYS” sequence (allowing 0 mismatch) in the Araport11 protein sequence database (www.arabidopsis.org) extracted 247 sequences with this motif. However, many of the identified proteins are not targeted to the secretory pathway and do not contain transmembrane domains (e.g. At1g06920, At1g08320, At2g21440), or the LPYS motif is found as part of a transmembrane domain (e.g. At1g18140, At1g60050, At5g01240) or part of the cleaved signal peptide (e.g. At1g65900, At3g50050) or present in the luminal domain (e.g. At5g10840). In these proteins LPYS has therefore a different function and different interacting proteins.

Same page, lines 414-415: instead of “Golgi complex” I would rather refer to all possible terms in the literature here (i.e. ERGIC/cisterna initiators/GECCO/Golgi remnants).

Our statement now includes the terms ERGIC, *cis* initiators, GECCO or Golgi remnants.

Page 14, lines 423-424: reconsider Golgi residency, when the HDEL tag successfully prevented the low MW band of MNS-XFP fusions on immunoblots!

The additional band is unspecific and generated during the extraction process. We have no indication that this band represents a transport intermediate or differently processed form of MNS3-mRFP-HDEL since this band disappears when using a different protein extraction protocol.

Same page: lines 428-430: ...or structural determinants, like the cytoskeleton or Golgi "glue" surrounding Golgi stacks in plant cells.

We deliberately did not mention any specific determinants since we only recently started to look for interacting candidates. For us, the term “unknown determinants” includes determinants such as Golgi tethers, matrix proteins, proteins involved in antero- and retrograde transport, regulatory proteins (Rabs), SNAREs, membrane lipids and others.

Same page, line 434-438: not all proteins, but those involved in N-glycan trimming/modification (Day et al. 2013). This is not an entirely NEW proposition, considering the earlier structural studies (ERGIC, cisternal initiators) and Ito et al. (2018), who recently coined “GECCO”.

We changed the term “Golgi-resident proteins” into “Golgi-resident glycosyltransferases and glycosidases”. We are aware that our proposition is not entirely new and hence we do cite other studies, such as those by Ito and colleagues. We are also now citing the structural studies by Day et al. (2013) and Donohoe et al. (2013).

Same page, line 442: After all, this interface is not so “ill defined”, considering the above mentioned structural studies of 2013.

We have changed the term “ill-defined” to “not so well-characterized”, since not so many structural studies are available up to now.

Same page, lines 444-450 meaning all proteins that travel through this cis-most compartment (ERGIC in other systems). Why would that be the case for ER residents – if reflecting passing through that interface, if not protecting them from further trimming by MNS4,5 in the ER?

Our model proposes that all/most ER-residents cycle through the MNS3-positive compartment for

removal of a single mannose residue. This could be a deliberate process to ensure/control N-glycan trimming for a yet unknown function (e.g. mannose trimming as a timer to control protein half-life of ER-residents) or is some kind of by-product of another process that requires the cycling through the MNS3 compartment.

Same page, lines 451-452: Really - interference with ERAD? You show that ER mislocalization of a GCS1-MNS3 fusion enhances ERAD of Bri1-5 by MNS4,5 – if this is not due to the CTS region of GCS1... (MNS3 plays no major role; Hüttner et al. 2014).

It is true that MNS3 does not play a role in the ERAD of misfolded BRI1-5 when correctly localized in the Golgi. That's exactly what we see in this work. Expression of Golgi-resident MNS3 in the *bri1-5* mutant has no effect on the *bri1-5* phenotype. Once localized in the ER, i.e. via the GCSI-CTS region, the *bri1-5* phenotype is strongly enhanced, due to the enhanced degradation of the misfolded BRI1-5 receptor. When we replaced the GCSI-CTS region of the UBQ10::GCSI-MNS3-GFP fusion protein with that of cytokinin oxidase/dehydrogenase CKX1 (Niemann et al., Plant Physiol 2018, doi:10.1104/pp.17.00925), which is an ER-resident type II single-pass membrane protein, and expressed the resulting UBQ10::CKX1-MNS3-GFP fusion protein in *bri1-5* plants, we observed an enhanced *bri1-5* growth defect similar to that observed for GCSI-MNS3-GFP or MNS3-L5A-GFP expressed in *bri1-5* plants (Suppl. Figure 12). In addition, *bri1-5* plants expressing GCSI-GnTI-GFP (containing the catalytic domain of GnTI instead of the MNS3 catalytic domain) did not show an enhanced phenotype and resembled *bri1-5* plants. This experiment emphasizes that the enhanced *bri1-5* growth defect results specifically from the presence of MNS3 in the ER and not from the GCSI-CTS region.

Page 15, line 471: cycles through (not to) the MNS3 compartment (...)

We have changed this sentence.

Same page, last para: Rather LPY is a newly identified “special signal” for accumulation in an intermediate layer (cis-most Golgi or ERGIC, neither entirely ER nor Golgi body yet), possibly due to interaction with a scaffold component (comment on cytoskeleton depolymerization drugs?)

It is indeed possible that the LPYS motif within the MNS3 cytoplasmic tail is a signal for its accumulation in an intermediate membrane structure that is left behind after Golgi disassembly and contains components such as Golgi matrix and tethering factors. We would like to point out that we did not compromise the cytoskeleton with any chemicals at any point, unless FRET-FLIM experiments and co-localization analyzes were performed (in this case, latrunculin B was used to inhibit actin-dependent movement of Golgi stacks). We have previously tested via coIP using GFP-Trap beads whether MNS3 interacts with AtCASP, a Golgi-localized tethering protein, and we could not find any interaction. At the moment, we do not have any indication that MNS3 interacts with a scaffold component leading to its retention on Golgi membranes or other structures such as the ERGIC, *cis* initiators, GECCO or Golgi remnants.

Material & Methods

Page 18-19, lines 587-589: What means “If necessary, leaf segments were treated with the actin-depolymerizing agent latrunculin B (Calbiochem, stock solution at 1 mM in dimethyl sulphoxide) at a

concentration 19 of 25 μ M for 45 to 60 min prior to image acquisition to inhibit Golgi movement.” This is an important information and should be mentioned in the context of BFA-resistant Golgi remnants.

We have included more explanation in the materials & methods section to help clarify that latrunculin B was exclusively used in the course of FRET-FLIM experiments and co-localization analyzes.

Figures

Please indicate MW sizes at the side of all immunoblots for better orientation.

We have now indicated the molecular weight sizes.

References

Please add those of the structural studies (Day et al. and Donohue et al. from 2013).

These studies have been added to the references.

REVIEWERS' COMMENTS:

Reviewer #1 (Remarks to the Author):

The authors have addressed my previous concerns.

Reviewer #2 (Remarks to the Author):

In this revision, the authors have invested great effort with additional experiments to address most of my previous comments, albeit the underlying mechanism remains elusive.

One minor comment: please check and make sure that their responses (text/discussion and related citations) have also been included in the revised MS accordingly (e.g. Supplemental Figure 9).

Reviewer #3 (Remarks to the Author):

I am satisfied by the responses to my comments from the authors. However, I would like to reply to a comment by Reviewer #1 who states "I feel that the results remain interesting for a subset of scientists which are in the plant glycobiology community." To the contrary, this paper and its findings should not be viewed as glycobiological esoterica. The modifications made on the oligosaccharide side chains of glycoproteins are signals used by the protein folding, quality control and export apparatus that handle nearly 30% of all proteins produced by a cell. The paper demonstrates that an important enzyme in the processing of the glycoprotein side chains is located at steady state in the cis-medial Golgi and not in the ER where it might mistakenly render well-folded proteins to destruction by the ERAD system. The authors also show that although the enzyme is located in the cis-medial Golgi it is not retrieved to the ER by BFA, a treatment that retrieves other cis-medial Golgi residents to the ER. They attribute this property to its retention in hitherto undefined punctate structures and to a cis-medial retention sequence on the enzyme, which is unique among other Golgi retention sequences known so far. The other reviewers then claim that a shortcoming of the paper is that the authors have not yet identified

the factor(s) to which this retention sequence interacts. That is important but beyond the scope of this paper. In all, this paper leaves some open questions, but it is a valuable contribution to our understanding of glycoprotein processing and the cellular disposition of the components of the processing machinery.

Reviewer #4 (Remarks to the Author):

Review report on Nature Communications manuscript NCOMMS-18-37261A

TITLE: A novel motif retains Arabidopsis ER- α -mannosidase I in the cis-Golgi and prevents enhanced glycoprotein ERAD, by: Jennifer Schoberer et al.

The authors have incorporated most points and comments raised by the Reviewers.

But, does MNS3 really cycles through the medial- (and trans-) Golgi cisternae, being firmly retained by its LPYS motif (even after BFA treatment)? Rather, the (mobile) N-glycan modification enzymes (glycosyltransferases and glycosidases) may do so – which would also fit the results, and emphasize the spatial separation of the cis- from the medial- (and trans-) Golgi cisternae, especially considering the perfect overlap of MNS3 with Syp31 (still shown as Suppl. Figure). Also, see/cite Chartre et al. (2009) J. Exp. Botany 60: 3157-65 for the L-containing ER export motif of Syp31.

The final model (Figure 9E) should be modified accordingly, with the *cis*-most cisterna compartment in which MNS3 resides (by interaction with an unknown component on the cytosolic side of the membrane), through which both ER residents/ERAD substrates (depicted on the left) and N-glycan modifying enzymes of the medial-Golgi like MNS1 and GnTI (depicted on the right) cycle. This would also result in (stationary) MNS3 with GNGN(XF)-terminated N-glycans.

Small things:

Page 15 (centre): Here, we propose that MNS3 ~~interferes~~ ENHANCES ~~with~~ the ERAD of misfolded glycoproteins like BRI1-5 when it is, (...)

Page 16: ERAD substrates likely also cycle ~~to~~ THROUGH this compartment, because (...).

Point-by-point response to reviewers' comments

ad Reviewer #2:

In this revision, the authors have invested great effort with additional experiments to address most of my previous comments, albeit the underlying mechanism remains elusive. One minor comment: please check and make sure that their responses (text/discussion and related citations) have also been included in the revised MS accordingly (e.g. Supplemental Figure 9).

As requested, we have added our responses (where applicable) and related citations to the revised manuscript.

ad Reviewer #4 (Remarks to the Author):

The authors have incorporated most points and comments raised by the Reviewers.

But, does MNS3 really cycle through the medial- (and trans-) Golgi cisternae, being firmly retained by its LPYS motif (even after BFA treatment)? Rather, the (mobile) N-glycan modification enzymes (glycosyltransferases and glycosidases) may do so – which would also fit the results, and emphasize the spatial separation of the cis- from the medial- (and trans-) Golgi cisternae, especially considering the perfect overlap of MNS3 with Syp31 (still shown as Suppl. Figure). Also, see/cite Chartre et al. (2009) J. Exp. Botany 60: 3157-65 for the L-containing ER export motif of Syp31.

The final model (Figure 9E) should be modified accordingly, with the *cis*-most cisterna compartment in which MNS3 resides (by interaction with an unknown component on the cytosolic side of the membrane), through which both ER residents/ERAD substrates (depicted on the left) and N-glycan modifying enzymes of the medial-Golgi like MNS1 and GnTI (depicted on the right) cycle. This would also result in (stationary) MNS3 with GNGN(XF)-terminated N-glycans.

We have highlighted the *cis*-most Golgi cisternae in our model as suggested by the reviewer.

We do think that MNS3 is a mobile protein that cycles through the Golgi stack and by doing so is processed by late-Golgi enzymes. We showed previously by fluorescence recovery after photobleaching (FRAP) that Golgi-resident glycosyltransferases and glycosidases such as GnTI, MII or GALT1 are not stably associated with the Golgi complex and instead undergo continuous movement in and out of Golgi stacks thereby receiving processing also by medial- and *trans*-Golgi N-glycan processing enzymes resulting in the generation of complex N-glycans (GnGnXF structures). For example, *N. tabacum* N-acetylglucosaminyltransferase I (GnTI) resides in *cis*/medial-Golgi cisternae and, like MNS3, predominantly carries GnGnXF structures (Schoberer et al., Traffic 2009, doi: 10.1111/j.1600-0854.2008.00841.x.; Plant J 2014, doi: 10.1111/tpj.12671). At the *trans*-Golgi, these proteins are then thought to be recycled back to their steady-state location in the Golgi, possibly involving a COPI-complex mediated retrieval mechanism. Preliminary FRAP experiments with MNS3-GFP or MNS3-CTS-GFP transiently expressed in *N. tabacum* leaves has indicated that most of the MNS3 protein population is mobile and recovers within minutes (fluorescence recovery rates and

mobile fractions are comparable to other previously studied Golgi-resident membrane proteins). In our opinion, MNS3 is then recycled from *trans*- to *cis*-Golgi cisternae via COPI. This is suggested by experiments where we transiently co-expressed MNS3 with RNAi constructs that silence endogenous δ COP and ϵ COP in *N. benthamiana* leaves. The silencing of these COPI subunits resulted in a shift of MNS3 to the vacuole probably due to a failure of efficiently recycling MNS3 from the *trans*-Golgi cisternae by COPI.